# Mechanisms of *Pseudomonas aeruginosa* resistance to type VI secretion system attacks

Alejandro Tejada-Arranz [1], Annika Plack[1], Minia Antelo-Varela [1,2], Andreas Kaczmarczyk [1], Alexander Klotz[1,3], Urs Jenal [1] & Marek Basler [1] ✉

The Type VI Secretion System (T6SS) is a molecular nanomachine that injects toxic effector proteins into the environment or neighboring cells, playing an important role in interbacterial competition and host antagonism during infection. *Pseudomonas aeruginosa* encodes three T6SSs. One of them, the H1-T6SS, delivers toxins in response to attacks mediated by the T6SS of aggressor bacteria, suggesting that *P. aeruginosa* can resist T6SS assaults. The mechanisms of resistance are poorly characterized. Here, we perform a CRISPRi screen to identify pathways involved in resistance to T6SS effectors of *Acinetobacter baylyi* ADP1 and *Vibrio cholerae* 2740-80. We show that members of the GacA/GacS regulon, such as the *mag* operon or *aas*, and GacA-independent factors, like the outer membrane protein OprF, confer resistance to different types of T6SS toxins. Interestingly, some of these T6SS protection mechanisms lead to higher antibiotic susceptibility, suggesting complex evolutionary links between T6SS and antibiotic resistance.

*P. aeruginosa* is a gram-negative, ubiquitous, opportunistic pathogen that can colonize open wounds and the respiratory airways, and is particularly problematic for patients suffering from cystic fibrosis[1]. *P. aeruginosa* possesses three Type VI Secretion Systems (T6SS), molecular nanoweapons that deliver toxic effectors to neighboring cells[2]. The first one, called H1-T6SS, plays a predominantly antibacterial role, while the H2- and H3-T6SSs are important for internalization and colonization of a eukaryotic host, although they also deliver transkingdom effectors that target both eukaryotic and bacterial cells[2–5]. The H3-T6SS also delivers effectors that help iron acquisition[6] and biofilm formation[7].

Generally, T6SSs are composed of a membrane complex and a baseplate that trigger the assembly of an Hcp tube that is loaded with a tip complex, composed of a VgrG trimer and PAAR proteins, as well as a set of effectors that can be fused to Hcp, VgrG or PAAR or can be loaded separately[8,9]. The membrane complex includes TssM and other structural components that are essential for T6SS function, span the inner and outer membranes and anchor the secretion system to the cell envelope[8]. The Hcp tube is surrounded by a sheath that, upon contraction, propels the Hcp tube as well as the tip complex and all loaded effectors to the extracellular space or into a neighboring cell. Bacteria are typically protected from their own effectors by a set of dedicated immunity proteins that are encoded adjacently to the effectors[10]. These immunity proteins are highly specific, with few examples of cross-protection to related effectors with similar mechanisms of action[11].

Interestingly, the H1-T6SS of *P. aeruginosa* was reported to respond to incoming attacks from neighboring bacteria, regardless of whether they are sister cells or members of a different species[12,13]. Specifically, the H1-T6SS comprises a sensor module, composed of TagQRST, that senses membrane damage and triggers the assembly of a retaliatory H1-T6SS through post-translational modifications[12]. Furthermore, it was proposed that this retaliation strategy is a poor evolutionary strategy, as it prevents *P. aeruginosa* from attacking first and forces it to resist incoming T6SS attacks[14]. Here, we hypothesize that *P. aeruginosa* must be equipped with specific mechanisms that protect against attacks with foreign T6SS effectors, at least until the H1-T6SS is able to strike back.

[1]Biozentrum, University of Basel, Basel, Switzerland. [2]Present address: Proteomics Unit, Gulbenkian Institute for Molecular Medicine, Lisbon, Portugal. [3]Present address: Department of Biosystems Science and Engineering, ETH Zurich, Basel, Switzerland. ✉e-mail: marek.basler@unibas.ch

*Acinetobacter baylyi* ADP1 and *Vibrio cholerae* 2740-80 are gram-negative environmental bacteria that each encode one antibacterial T6SS cluster with multiple effectors that are constitutively expressed and thus highly active in vitro. *A. baylyi* ADP1 WT contains four described effectors[15]: Tpe1 (a predicted metallopeptidase without identified antimicrobial activity), Tae1 (a PG-targeting effector), Tse2 (of unknown mechanism of action) and Tle1 (a lipase). *V. cholerae* 2740-80 WT, a representative strain for AAA effector-type, non-toxigenic environmental *V. cholerae*[16], also has four effectors[17]: TseH and VgrG3 (two PG-targeting effectors), TseL (a lipase) and VasX (a pore-forming toxin). Thus, they have different types of effectors, and the resistance mechanisms of *P. aeruginosa* against these two organisms might differ.

In this work, we competed H1-T6SS-deficient *P. aeruginosa* PAO1 carrying a CRISPRi plasmid library targeting all *P. aeruginosa* genes with T6SS-positive or –negative *A. baylyi* ADP1 or *V. cholerae* 2740-80. Deep sequencing of the plasmid library revealed genes required to resist incoming T6SS attacks. Interestingly, most of these genes belong to the regulon of the GacS/GacA two-component system (TCS), including the previously identified *arc1* and *arc3* clusters and *pa3267*, *aas*[18], as well as the *mag* operon. The outer membrane protein (OMP) OprF also contributes to T6SS resistance. Our data suggest that these pathways are general protection mechanisms against multiple effectors and thus differ from specific immunity proteins. In addition, we identified a link between these protection mechanisms and antibiotic resistance.

## Results

### *P. aeruginosa* resists T6SS attacks in the absence of its H1-T6SS

The H1-T6SS of *P. aeruginosa* can promptly and efficiently retaliate to incoming T6SS attacks, eliminating the competing bacterium before it can efficiently kill *P. aeruginosa*[12,14,19]. Therefore, we constructed a mutant lacking *tssM1*, a structural membrane component of the H1-T6SS, that is unable to assemble its H1-T6SS and retaliate to incoming attacks. We observed that the H1-T6SS is indeed important to efficiently resist attacks from *A. baylyi* ADP1 (Fig. 1A and Supplementary Fig. 1A), but not from *V. cholerae* 2740-80 (Fig. 1B and Supplementary Fig. 1B) in a 10:1 attacker:prey ratio. The H2- and H3-T6SSs of *P. aeruginosa* play no role in resistance against T6SS attacks (Fig. 1A, B and Supplementary Fig. 1A, B). Under the tested conditions, where *P. aeruginosa* is outnumbered by the attacking species (10:1 attacker: prey ratio), no significant difference in attacker recovery was observed (Supplementary Fig. 1A, B). Nevertheless, H1-T6SS-dependent killing of *A. baylyi* and *V. cholerae* can be observed when competed against *P. aeruginosa* in a 1:1 ratio or when *P. aeruginosa* outnumbers the other species in a 10:1 ratio (Supplementary Fig. 1C–F, where *P. aeruginosa* is the attacker and *A. baylyi* and *V. cholerae* the prey).

Furthermore, although the *P. aeruginosa* Δ*tssM1* strain is more susceptible than its parental strain to attacks from the T6SS of *A. baylyi* ADP1, it is still far more resistant than *E. coli* (Figs. 1A, C and Supplementary Fig. 1A, G), as evidenced by the almost complete elimination of *E. coli* when competed in a 10:1 attacker: prey ratio and the strong killing observed in a 1:1 ratio, in a manner that is dependent on the presence of the known *A. baylyi* ADP1 and *V. cholerae* 2740-80 effectors (Fig. 1C). This suggests that *P. aeruginosa* possesses protective mechanisms against T6SS attacks.

### Identification of T6SS-protective pathways

To identify new T6SS-protective pathways, we introduced the dCas9 protein from *Streptococcus pasteurianus* into *P. aeruginosa* Δ*tssM1* strain and performed a CRISPRi screen for genes important for T6SS protection using a sgRNA library similarly to others[20,21]. Specifically, we competed this T6SS-deficient *P. aeruginosa* strain carrying the sgRNA plasmid library with T6SS-active or inactive mutants (lacking *tssM*) of *A. baylyi* ADP1 or *V. cholerae* 2740-80. We performed two rounds of 3 h

long incubations of predator and prey at 1:1 (for *A. baylyi*) and 10:1 (for *V. cholerae*) ratios (Fig. 2A). The sgRNA loci of the recovered *P. aeruginosa* cells were PCR-amplified (before competition and after the first and second rounds of competition), and the amplicons were sequenced by Illumina sequencing. Samples from competition with T6SS-active or inactive strains were compared to identify the pathways that, when knocked down, lead to higher sensitivity to T6SS attacks.

After one and two rounds of competition, plasmids encoding sgRNAs targeting a number of *P. aeruginosa* genes were depleted (Fig. 2B, C and Supplementary Fig. 2A, B). The same sgRNAs were identified in both cases, although differences were more pronounced after two rounds of competition. This included known T6SS resistance-related genes such as *gacA*, *gacS*, *aas*, *arc1A* or *arc3A*[18] as well as novel genes like the *mag* operon or *oprF*.

### GacA is a key regulator involved in T6SS resistance

Interestingly, many of the genes that we identified were reported to be under the control of the GacA/GacS TCS, including *arc1A*, *arc3A* and *aas*[18]. To determine whether the other genes that we identified are also part of the regulon of the GacA/S TCS, we compared the proteome of a Δ*gacA* strain to that of its parental strain. We found that the members of the *mag* operon are also downregulated in the Δ*gacA* strain (Fig. 2D, and Supplementary Data 1), together with components of the H1- and H2-T6SS, whereas *oprF* is not differentially expressed in that strain.

Next, we assessed whether a Δ*gacA* strain was able to resist T6SS attacks from *A. baylyi* ADP1 and *V. cholerae* 2740-80. Indeed, the Δ*gacA* strain was 100-fold more susceptible to T6SS attacks from *A. baylyi* ADP1 and *V. cholerae* 2740-80 than a strain lacking the H1-T6SS (Fig. 2E and Supplementary Fig. 2C, D), indicating that GacA-regulated genes other than the H1-T6SS are required for T6SS resistance.

To get more insights into the mechanisms of resistance, we competed the *P. aeruginosa* Δ*gacA* strain with *A. baylyi* strains lacking individual effectors (Fig. 2F and Supplementary Fig. 2E). We previously showed that these *A. baylyi* strains are able to fire their T6SS as efficiently as the parental strain[15]. Interestingly, the Δ*gacA* strain is killed similarly when single effectors are absent, although Tse2 might play the predominant role, as an *A. baylyi* strain lacking *tse2* kills *P. aeruginosa* less efficiently (Fig. 2F).

Furthermore, we also competed *P. aeruginosa* Δ*gacA* with *V. cholerae* 2740-80 strains encoding individually inactivated effectors through point mutations that were previously reported[13,22], and we found that the inactivation of each effector leads to similar levels of killing of the Δ*gacA* strain (Fig. 2G and Supplementary Fig. 2F). Taken together, our results suggest that the GacA regulon contains genes that contribute to the resistance to different effector classes, in addition to lipase effectors as previously reported[18].

### The *mag* operon is important for resistance against PG-targeting effectors

Next, we assessed how individual genes from the GacA regulon contribute to T6SS resistance, and in particular the *mag* operon that was identified in our CRISPRi screen (Fig. 2B). In *P. aeruginosa*, the *mag* operon is composed of six genes, *magABCDEF*. MagD displays weak homology with type II α2-macroglobulin[23], homologs of eukaryotic α-macroglobulins, whereas the rest of the genes of the operon are important for the folding and stability of MagD[24]. Eukaryotic α2-macroglobulins are innate immunity proteins that protect against proteases by a trapping mechanism[25–27]. However, bacterial type II α2-macroglobulins lack key residues that conform the active site of eukaryotic α2-macroglobulins[24], and their function is thus unclear.

We constructed a strain lacking both *tssM1* and *magD*, and competed it against different mutants of *V. cholerae* 2740-80. We performed quantitative killing assays and found that MagD is indeed important for resistance to T6SS attacks and that this phenotype can

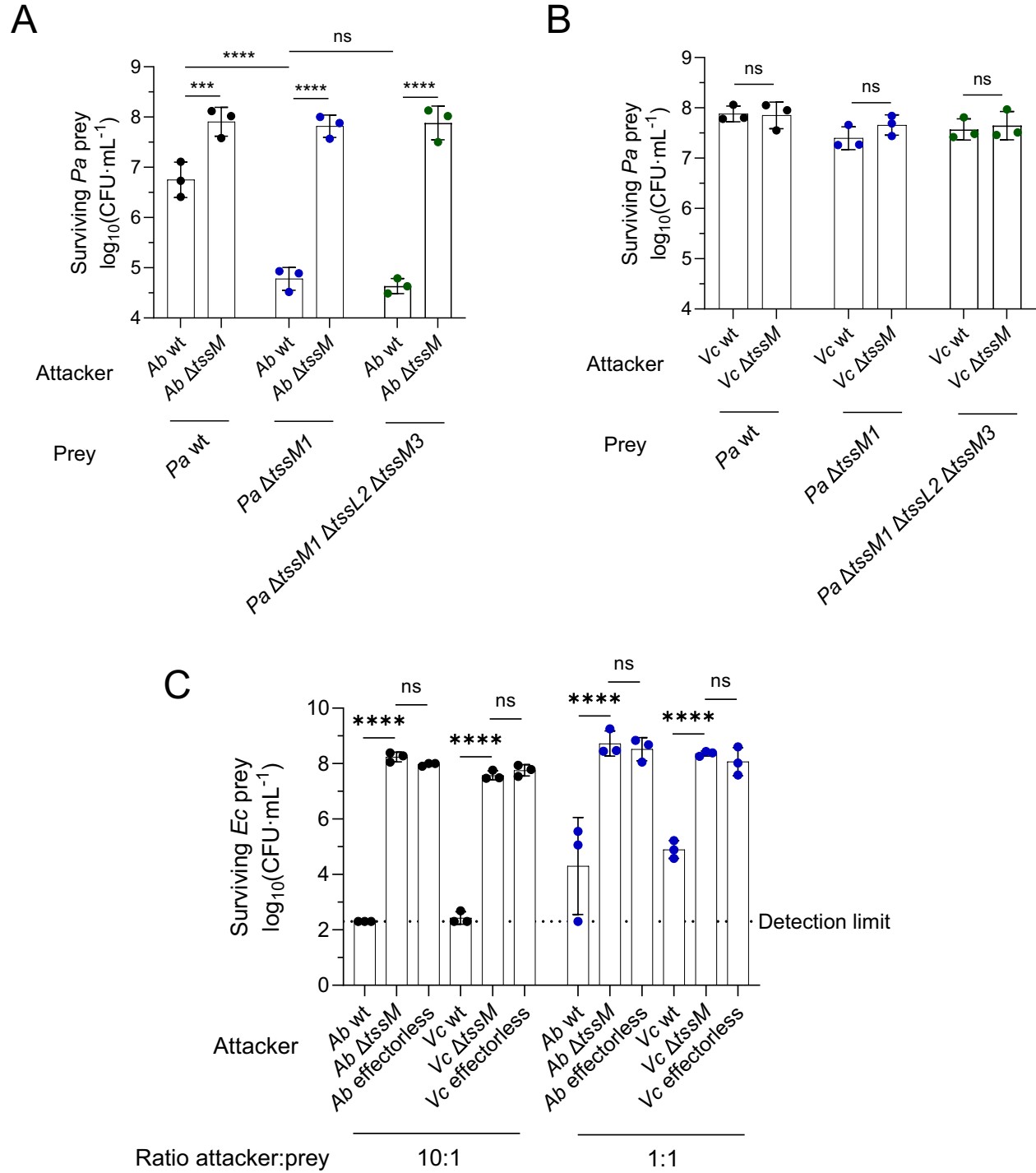

**Fig. 1 | _P. aeruginosa_ (_Pa_) is more resistant to T6SS from _A. baylyi_ (_Ab_) ADP1 and _V. cholerae_ (_Vc_) 2740-80 than _E. coli_ (_Ec_).** CFU counts showing the survival of _P. aeruginosa_ when competed against _A. baylyi_ ADP1 (_Ab_) (**A**) and _V. cholerae_ 2740-80 (_Vc_) (**B**) strains that have or do not have an active H1-, H2- and H3-T6SS, in a 10:1 attacker:prey ratio where _P. aeruginosa_ is the prey. **C** CFU counts showing the survival of _E. coli_ (_Ec_) when competed against _A. baylyi_ and _V. cholerae_ in 10:1 and 1:1 attacker:prey ratios where _E. coli_ is the prey, when _A. baylyi_ and _V. cholerae_ have or do not carry any known

active T6SS effectors (effector deletions for _A. baylyi_ and effector-inactivating mutations for _V. cholerae_). Source data are provided as a Source Data file. The bars indicate the mean across three independent biological replicates, the error bars indicate the standard deviation. Each biological replicate represents one data point. Data were analyzed with ordinary two-way ANOVA and Sidak's correction for multiple comparisons. Statistically significant adjusted p-values are indicated with asterisks and are (left to right) 0.0006*** and <0.0001****. ns: non-significant. The datapoints are colored according to the _Pa_ strain used.

be complemented (Fig. 3A and Supplementary Fig. 3A). Furthermore, MagD protects mainly against the PG-targeting effector VgrG3 (Fig. 3B and Supplementary Fig. 3B). To confirm this result, we also performed a CPRG-based lysis assay[15]. CPRG is a colorimetric substrate that cannot

penetrate bacterial membranes and can thus only be converted by the LacZ enzyme upon lysis of LacZ-expressing _P. aeruginosa_, acting as a proxy for lysis. This assay also indicated a VgrG3-dependent lysis of a Δ_magD_ mutant (Fig. 3C and Supplementary Fig. 3C).

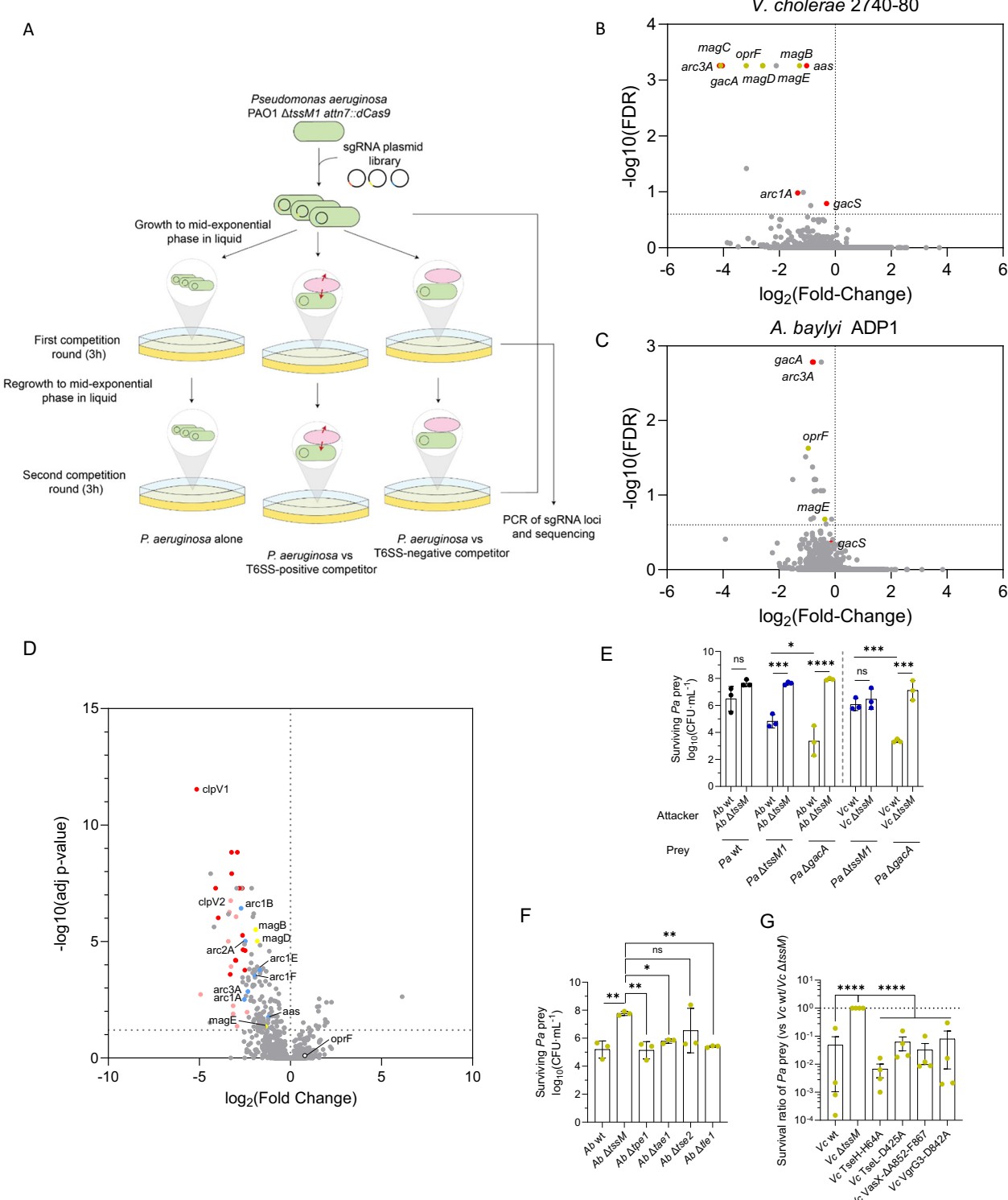

Additionally, we tested whether *magD* confers resistance against single effectors of *A. baylyi* by competition with strains expressing single effectors. We found that *magD* also confers a limited protection against the PG-targeting Tae1 effector of *A. baylyi* (Fig. 3D and Supplementary Fig. 3D), and not against other effector classes (Supplementary Fig. 3E, F). Thus, MagD encoded by the *mag* operon protects against certain PG-targeting effectors from both *V. cholerae* 2740-80 and *A. baylyi* ADP1.

## Arc1A, Arc3A and Aas contribute to resistance against lipase effectors

Arc1A and Arc3A were previously shown to be important for resistance against the T6SS of *B. thailandensis*. Specifically, Arc3A protects against lipase effectors, and the effector class that Arc1A protects against was not clearly elucidated[18]. PA3267 (Aas) was also previously identified in a transposon screen, but no clear role in T6SS resistance was attributed to it, although it was hypothesized that it is relevant for

**Fig. 2 | CRISPRi screen for *P. aeruginosa* knockdowns that are sensitive to T6SS attacks. A** Schematic of the experimental design. *P. aeruginosa* lacking *tssM1* and carrying the sgRNA library was competed against T6SS-positive or –negative *A. baylyi* and *V. cholerae* for 1 or 2 rounds of competition lasting 3 h each. Bacteria were grown to mid-exponential phase before each competition round. **B** Negative-selection Volcano plot showing genes important for *P. aeruginosa* resistance to T6SS from *V. cholerae*, after competition in a 10:1 attacker:prey ratio. In red are previously identified genes and in yellow genes identified in this study. **C** Negative-selection Volcano plot showing genes encoding proteins important for *P. aeruginosa* resistance to T6SS from *A. baylyi*, after competition in a 1:1 attacker:prey ratio. In red are previously identified genes and in yellow genes identified in this study. **D** Volcano plot showing proteins that are differentially regulated in a Δ*gacA* strain compared to wild-type *P. aeruginosa*. In red are components of the H1-T6SS, in pink components of the H2-T6SS, in blue previously identified members of the GacA regulon that are linked to resistance to T6SS attacks of *B. thailandensis*[18] and in yellow the identified hits from the *mag* operon. *oprF* is also highlighted as a non-differentially regulated gene. Relative protein abundances and pairwise comparisons were calculated using the MSstats package[58] with default settings selected. **E** CFU counts showing the survival of different *P. aeruginosa* (*Pa*) prey strains when competed against T6SS-positive or negative *A. baylyi* (*Ab*) and *V. cholerae* (*Vc*) in a 10:1 attacker to prey ratio. Data were analyzed with ordinary two-way ANOVA and Sidak's correction for multiple comparisons. Statistically significant adjusted p-values are indicated with asterisks and are *0.0285, ***0.0004, 0.0008, 0.0001 (left to right), and ****<0.0001. ns: non-significant. The datapoints are colored according to the *Pa* strain used. **F** CFU counts showing the survival of prey *P. aeruginosa* Δ*gacA* when competed against *A. baylyi* lacking different T6SS effectors in a 10:1 attacker to prey ratio. Data were analyzed with ordinary one-way ANOVA and Dunnett's correction for multiple comparisons. Statistically significant adjusted p-values are indicated with asterisks and are *0.0272 and **0.0048, 0.0041, 0.0089 (left to right). ns: non-significant. **G** Percentage of survival of prey *P. aeruginosa* Δ*gacA* when competed against *V. cholerae* carrying inactivated versions of different T6SS effectors, compared to *V. cholerae* Δ*tssM*, in a 10:1 attacker to prey ratio. Data were analyzed with ordinary one-way ANOVA and Dunnett's correction for multiple comparisons. Statistically significant adjusted p-values are indicated with asterisks and are ****<0.0001. For all panels, source data are provided as a Source Data file. **E–G** The bars indicate the mean across three independent biological replicates, the error bars indicate the standard deviation. Each biological replicate represents one data point.

protection against lipases[18]. Our screen identified *arc1A*, *arc3A* and *aas* as important for T6SS resistance. Aas is homologous to the lysophospholipid transporter LplT and the Aas acyltransferase from *E. coli*, where it was shown to be required for defense against phospholipase attacks[28]. We found that a *P. aeruginosa* mutant lacking *tssM1* and either *arc1A*, *arc3A* or *aas* is indeed more sensitive to T6SS attacks from *V. cholerae* 2740-80 WT (Fig. 4A and Supplementary Fig. 3G). The Δ*tssM1* Δ*arc3A* displays the strongest defect. A strain lacking all four genes is as sensitive to attacks as the Δ*tssM1* Δ*arc3A* strain, suggesting an auxiliary role of *arc1A* and *aas* in this process. This sensitivity is reduced when *V. cholerae* 2740-80 lacks an active TseL effector (Fig. 4A and Supplementary Fig. 3G). This suggests that these genes are important for protection against lipases, and likely also contribute to resistance against other effector classes.

Similarly, *P. aeruginosa* strains lacking *tssM1* and either *arc1A* or *arc3A* are also more sensitive than *P. aeruginosa* Δ*tssM1* to attacks from *A. baylyi* ADP1 that carries only its lipase Tle1 effector (Fig. 4B and Supplementary Fig. 3H). The strain lacking all four genes (*tssM1*, *arc1A*, *arc3A* and *aas*) was as sensitive to either Tae1 or Tse2 from *A. baylyi* ADP1 as *P. aeruginosa* Δ*tssM1* (Supplementary Fig. 3I, J). We conclude that *arc1A*, *arc3A* and *aas* are important for resistance to lipase effectors, confirming previous observations about *arc3A*[18], providing evidence for the previously suggested role of *aas*[18] and identifying one of the types of effectors that *arc1A* can protect against.

## OprF is required for T6SS resistance and is genetically linked to GacA/S

OprF is the main outer membrane porin of *P. aeruginosa*, and it has been reported to perform many different roles, including channeling small solutes, maintenance of outer membrane integrity, binding and adhesion, biofilm formation, or outer membrane vesicle biogenesis[29]. OprF is folded in two conformers: a minor population (around 5%) that is folded into a one-domain structure with a channel that acts as a porin; and a major population (around 95%) that is folded into two domains, including a PG-anchoring domain. It was suggested that this conformation is responsible for anchoring of the OM to the PG[30,31], and that a Δ*oprF* strain displays defects in the OM[32,33]. To understand how sgRNA-mediated inhibition of *oprF* expression resulted in lower resistance to T6SS attacks, we constructed a *P. aeruginosa* Δ*tssM1* Δ*oprF* strain and competed it against *V. cholerae* 2740-80 and *A. baylyi* ADP1 attackers. We show that when the *V. cholerae* 2740-80 attacker strain carries a full effector set, the Δ*oprF* strain is killed (Fig. 5A and Supplementary Fig. 4A). Inactivation of single or multiple effectors indicates that VgrG3 is the main driver of this killing. When *A. baylyi* ADP1 is used as attacker, *P. aeruginosa* Δ*tssM1* Δ*oprF* is also more

susceptible than its parental strain, independently of the presence of any individual effector, indicating that no single effector type is responsible for the killing of *P. aeruginosa* lacking *oprF* (Fig. 5B and Supplementary Fig. 4B). Importantly, the Δ*oprF* strain displays a growth defect in LB, which is exacerbated in LB without salt (Supplementary Fig. 4C), suggesting that the strain is sensitive to osmotic shock as previously shown[32].

Furthermore, complementation in *trans* with a wild-type copy of *oprF* in a pPSV35 plasmid fully restores both growth and resistance to T6SS attacks (Fig. 5C and Supplementary Fig. 4D, E). The R296E *oprF* variant, that was previously shown to be unable to bind PG in *E. coli*[34,35], and fails to complement morphology defects derived from envelope instability (Supplementary Fig. 4E) restores protection against T6SS attacks to the same extent (Fig. 5C and Supplementary Fig. 4F), suggesting that the contribution of the OM anchoring function of OprF to T6SS resistance may be minimal.

Interestingly, we observed different colony sizes when the *P. aeruginosa* Δ*tssM1* Δ*oprF* strain was grown in LB without salt, suggesting the appearance of suppressors (Fig. 6A). We isolated four of these clones and analyzed their growth, cell morphology and ability to compete with *V. cholerae* 2740-80. These suppressors displayed wild-type-like growth on LB without salt (Fig. 6B), as well as wild-type cellular morphology (Fig. 6C), as shown by the absence of cell ghosts that are frequent in the Δ*tssM1* Δ*oprF* strain. However, the suppressors were still susceptible to T6SS attacks from *V. cholerae* 2740-80 at a 1:1 ratio (Fig. 6D and Supplementary Fig. 4G).

We performed whole genome sequencing of these suppressors and found that all of them carry the same inactivating 431 bp deletion within the *gacS* gene, suggesting that the GacA/S is inactive in these strains. We validated this by deletion of *gacA* in a Δ*oprF* strain, which restores the growth defect of this strain in LB without salt (Fig. 6E). Thus, GacA/S is involved in causing the growth and morphology defect of the Δ*oprF* strain, which explains why these suppressors are still sensitive to T6SS attacks, as they lack a functional GacA/S that is important for resistance to such attacks (Fig. 2). Our results show that there is a genetic link between *oprF* and the GacA/S TCS.

## T6SS resistance mechanisms are effector-type specific

While the previous study from Ting et al.[18]. identified three operons, *arc1*, *arc2* and *arc3*, involved in protection against T6SS from *Burkholderia thailandensis*, only two of them, *arc1* and *arc3* appeared in our CRISPRi screen. Moreover, the two further resistance factors that we identified, the *mag* operon and *oprF*, were not identified in the previous transposon screen. To verify whether these mechanisms are

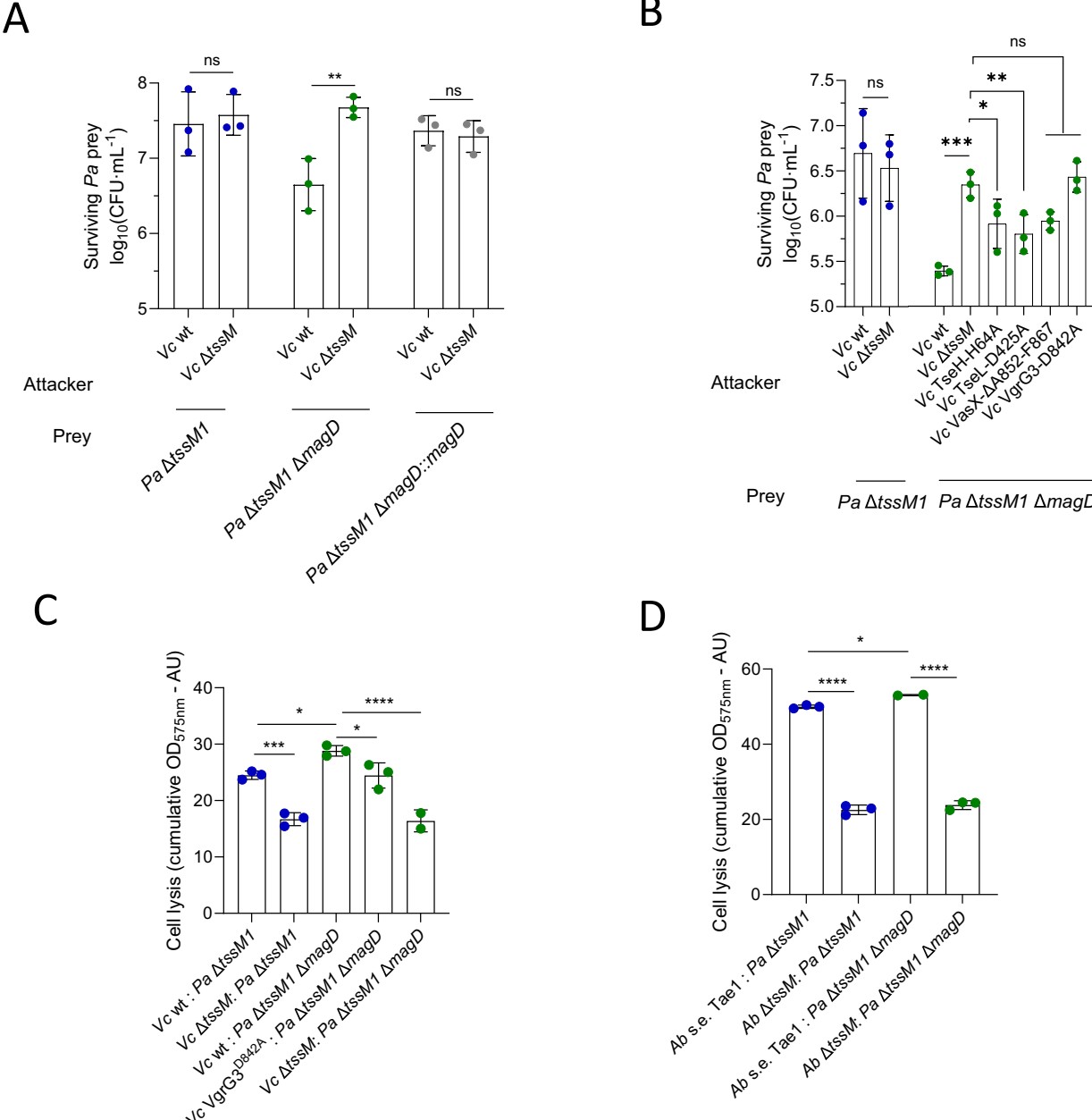

**Fig. 3 | MagD is important for resistance against PG-targeting effectors. A** CFU counts showing the survival of prey *P. aeruginosa* (*Pa*) strains when competed against *V. cholerae* (*Vc*) that have or do not have an active T6SS, in a 10:1 attacker to prey ratio. Data were analyzed with ordinary two-way ANOVA and Sidak's correction for multiple comparisons. Statistically significant adjusted p-values are indicated with asterisks and are **0.0024. **B** CFU counts showing the survival of different prey *P. aeruginosa* strains when competed against *V. cholerae* carrying different inactivated effectors, in a 10:1 attacker to prey ratio. Data were analyzed with ordinary one-way ANOVA and Dunnett's correction for multiple comparisons. Statistically significant adjusted p-values are indicated with asterisks and are *0.0397, **0.0098 and ****<0.0001. **C** Cumulative OD$_{575nm}$ due to the red conversion of CPRG dye (proxy for cell lysis of prey LacZ+ *P. aeruginosa*) over a period of 10 h at 30 °C when competed against *V. cholerae* in a 2:1 attacker to prey ratio. Data

were analyzed with ordinary one-way ANOVA and Dunnett's correction for multiple comparisons. Statistically significant adjusted p-values are indicated with asterisks and are *0.0223, 0.0210 (left to right), ***0.0004 and ****<0.0001. **D** Cumulative OD$_{575nm}$ due to the red conversion of CPRG dye (proxy for cell lysis of prey LacZ+ *P. aeruginosa*) over a period of 10 h at 30 °C when competed against *A. baylyi* (*Ab*) carrying only its PG-targeting effector, Tae1, in a 2:1 attacker to prey ratio. Data were analyzed with ordinary one-way ANOVA and Dunnett's correction for multiple comparisons. Statistically significant adjusted p-values are indicated with asterisks and are *0.0279 and ****<0.0001. For all panels, source data are provided as a Source Data file. The bars indicate the mean across three independent biological replicates, the error bars indicate the standard deviation. Each biological replicate represents one data point. ns: non-significant. The datapoints are colored according to the *Pa* strain used.

important for resistance to other organisms, we also tested the susceptibility of *P. aeruginosa* Δ*tssM1*, Δ*gacA*, Δ*tssM1* Δ*magD*, Δ*tssM1* Δ*aas* and Δ*tssM1* Δ*oprF* to *B. thailandensis* attacks (Fig. 7 and Supplementary Fig. 5). Although *magD* is dispensable for resistance to the T6SS of *B. thailandensis*, both *gacA* and *oprF* are important, albeit to a lesser

extent than in the case of *A. baylyi* ADP1 or *V. cholerae* 2740-80. Thus, some T6SS-protective mechanisms, such as those in the GacA regulon, are specific to certain effector types that may be carried by different strains, and others, like OprF-mediated mechanisms, appear to be generic protection mechanisms.

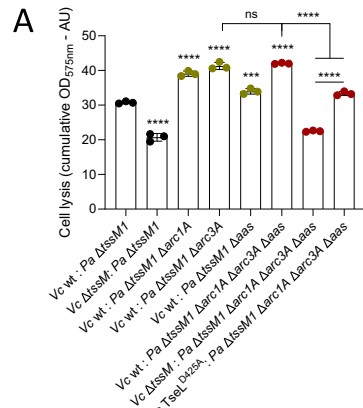

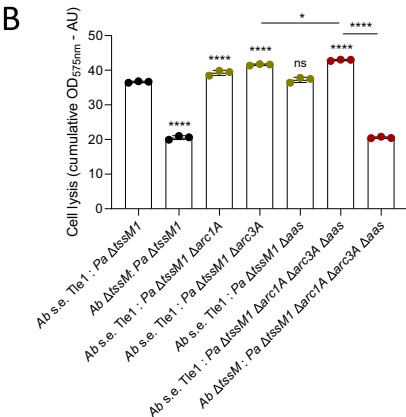

**Fig. 4 | Arc1A, Arc3A and Aas are important for resistance to lipase effectors.**
Cumulative $OD_{575nm}$ due to the red conversion of CPRG dye (proxy for cell lysis of prey LacZ+ *P. aeruginosa* (*Pa*)) over a period of 10 h at 30 °C when competed against **A** *V. cholerae* (*Vc*) and **B** *A. baylyi* (*Ab*) carrying only its lipase effector, Tle1, in a 2:1 attacker to prey ratio. Statistically significant differences are indicated with respect to the competition against *P. aeruginosa* Δ*tssM1* (parental) in the first bar. Data were analyzed with ordinary one-way ANOVA and Sidak's correction for multiple comparisons. Statistically significant adjusted p-values are indicated with asterisks and are for **A** ***0.0003 and ****<0.0001, and for **B** *0.0299 and ****<0.0001. ns: non-significant. Source data are provided as a Source Data file. The bars indicate the mean across three independent biological replicates, the error bars indicate the standard deviation. Each biological replicate represents one data point. The datapoints are colored according to the *Pa* strain used.

## T6SS resistance strategies can affect antibiotic susceptibility

To determine whether T6SS protection mechanisms might be important for antibiotic resistance, we explored the susceptibility of T6SS-sensitive strains to 30 antibiotics from different classes, including inhibitors of translation, cell wall synthesis, folate synthesis, membrane integrity, DNA replication and transcription, among others. We performed growth curves of the Δ*gacA*, Δ*tssM1*, Δ*tssM1* Δ*arc1A*, Δ*arc3A* Δ*aas*, Δ*magD* and the Δ*oprF* strains in the presence of antibiotics and compared the cumulative $OD_{600 nm}$ for each condition, relative to wild-type and to the untreated condition (Fig. 8A and Supplementary Figs. 6–8). Interestingly, all tested strains displayed altered antibiotic susceptibility profiles compared to wild-type. Specifically, the Δ*tssM1* strain displays increased growth in the presence of nadifloxacin, which is extended to other fluoroquinolones by the additional deletion of *arc1A, arc3A* and *aas* (Fig. 8A, B). These effects were exacerbated in the Δ*gacA* strain that does not express these and other genes that may be implicated in this phenotype. *P. aeruginosa* Δ*magD* is more resistant to chlorhexidine, an antimicrobial compound targeting the bacterial cell membrane (Figs. 8A, C). Lastly, the *P. aeruginosa* Δ*oprF* strain is more resistant to antibiotics like the membrane-targeting colistin (Figs. 8A, D) and translation-targeting gentamicin, and more sensitive to the translation inhibitor tetracycline (Figs. 8A, E) and beta-lactams, such as piperacillin or azlocillin (Fig. 8A). This suggests that T6SS resistance mechanisms alter the antibiotic susceptibility profile of *P. aeruginosa*.

## Discussion

*P. aeruginosa* is able to promptly sense and retaliate to T6SS attacks from other species[12]. However, the mechanisms that allow *P. aeruginosa* to withstand the initial attack (Fig. 1) are not well understood. Only one recent report[18] identified three operons involved in resistance to T6SS attacks from *B. thailandensis*, called *arc1, arc2* and *arc3*, and determined that *arc3* is particularly important to resist the delivery of lipase effectors, likely through the detoxification of lysophospholipids that are generated as a result of the activity of these lipases. However, the effector types that *arc1* and *arc2* confer protection against, and whether there are other mechanisms involved in the resistance to other effector types, are not known. Here, we performed a CRISPRi screen to identify further factors involved in T6SS resistance in *P. aeruginosa*, using the laboratory models *A. baylyi* ADP1 and *V. cholerae* 2740-80 (AAA effector-type[16] strain) as aggressor species.

These two strains contain different effector sets[15,17]: both of them have at least one lipase (Tle1 in *A. baylyi* and TseL in *V. cholerae*) and at least one PG-targeting effector (Tae1 in *A. baylyi* and VgrG3 and TseH in *V. cholerae*); *A. baylyi* encodes two other effectors of unknown activity (Tpe1 – a predicted metallopeptidase - and Tse2); and *V. cholerae* encodes one membrane-targeting pore-forming effector VasX.

In our screen, we identified known factors, such as the *arc1* and *arc3* clusters, as well as *gacS* and *gacA* of the GacA/S TCS that regulates the expression of said clusters[18] in addition to the expression of the H1- and H2-T6SS[36,37] (Fig. 2B, C, E, G). We also identified further factors that contribute to resistance, the *mag* operon, that is also regulated by GacA, underscoring the importance of this transcriptional regulator for the survival of *P. aeruginosa* in polymicrobial communities. Furthermore, we found other factors that are not regulated by GacA (Fig. 2D and Fig. 5), such as the OMP OprF.

Through quantitative killing assays and CPRG-based lysis assays, we could validate that these factors are indeed involved in protection against T6SS attacks (Figs. 2–5). In agreement with Ting et al.[18], we also observed that *arc3* is important specifically to resist to lipases (Fig. 4). Here, we also observe that *arc1A* as well as *aas* are required for full resistance against the lipases from both *V. cholerae* 2740-80 and *A. baylyi* ADP1 (Fig. 4). Previously, *aas* was hypothesized to protect against lipases, however, *arc1* role in protection against *B. thailandensis* effectors remained unclear[18]. This suggests that the specific catalytic mechanism of the lipases in question might be relevant for the mode of action of the corresponding resistance mechanisms.

Our results also show that the GacA regulon is necessary for the resistance against effector classes other than lipases (Fig. 2F, G). We found that the *mag* operon, that is regulated by GacA, is particularly important to resist against certain PG-targeting effectors, such as VgrG3 from *V. cholerae* 2740-80 and Tae1 from *A. baylyi* ADP1 (Fig. 3). Interestingly, the central gene of this operon, *magD*, displays homology with human α2-macroglobulins[23], innate immunity factors that defend against proteases through a trapping mechanism[25–27]. However, the catalytic residues of this protein family are not conserved in bacterial type II α2-macroglobulins that *magD* belongs to[24], raising questions about the true function of this protein family in bacteria. Here, we found that MagD contributes to protection against some T6SS PG-targeting effectors (VgrG3 and Tae1, but not TseH), although the exact molecular mechanism remains to be elucidated. Previous studies suggest that TseH is a NlpC/P60 family cysteine endopeptidase[38],

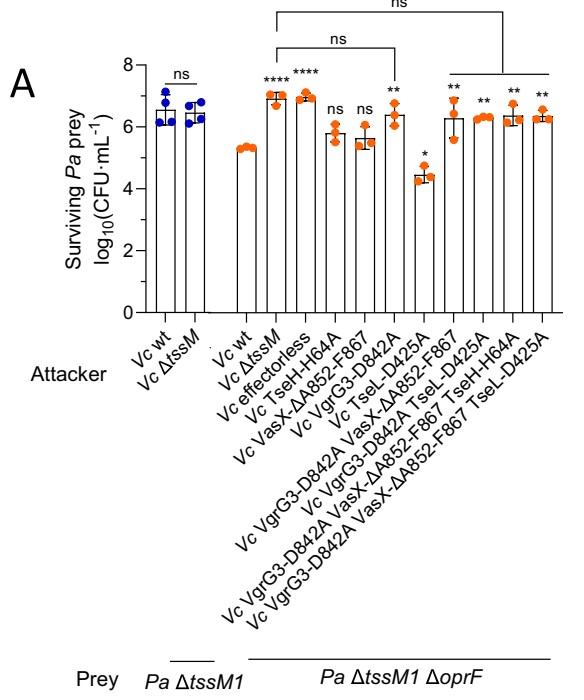

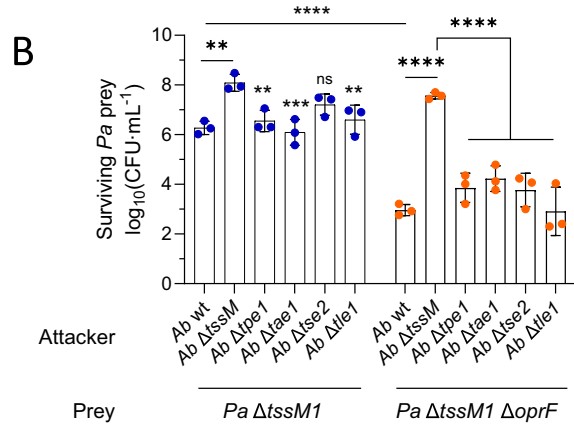

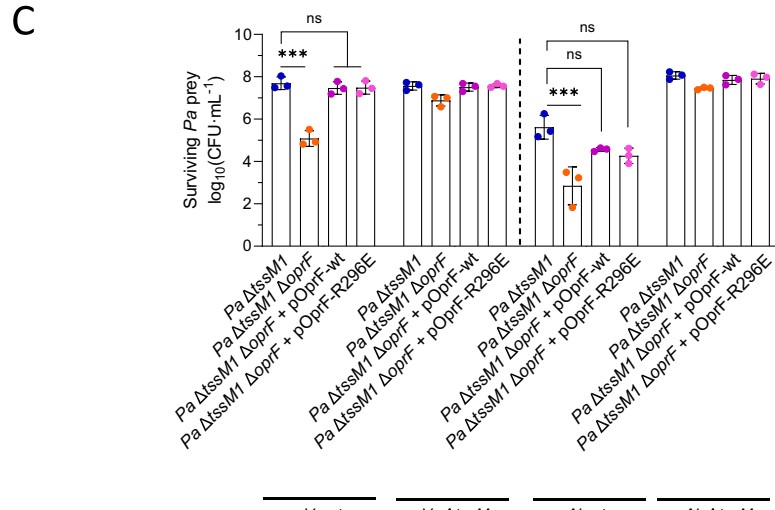

**Fig. 5 | OprF is important for T6SS resistance. A** CFU counts showing the survival of prey *P. aeruginosa* (*Pa*) strains when competed against *V. cholerae* (*Vc*) strains carrying wild-type or inactivated T6SS effectors in a 10:1 attacker to prey ratio. Statistically significant differences are indicated with respect to the competition with *V. cholerae* WT unless otherwise depicted. Data were analyzed with ordinary one-way ANOVA and Dunnett's correction for multiple comparisons. Statistically significant adjusted p-values are indicated with asterisks and are *0.0166, **0.0025, 0.0071, 0.0064, 0.0031, 0.0037 (left to right) and ****<0.0001. **B** CFU counts showing the survival of prey *P. aeruginosa* strains when competed against *A. baylyi* (*Ab*) strains lacking single T6SS effectors in a 10:1 attacker to prey ratio. Statistically significant differences are indicated with respect to the competition with *A. baylyi* Δ*tssM* unless otherwise depicted. Data were analyzed with ordinary two-way

ANOVA and Sidak's correction for multiple comparisons. Statistically significant adjusted p-values are indicated with asterisks and are **0.0012, 0.0060, 0.0078 (left to right), ***0.0004 and ****<0.0001. **C** CFU counts showing the survival of prey *P. aeruginosa* strains complemented with wild-type or mutant variants of *oprF* when competed against *V. cholerae* and *A. baylyi* in a 10:1 attacker to prey ratio. Data were analyzed with ordinary one-way ANOVA and Dunnett's correction for multiple comparisons. Statistically significant adjusted p-values are indicated with asterisks and are ***0.0002, 0.0009 (left to right) and ****<0.0001. For all panels, source data are provided as a Source Data file. The bars indicate the mean across three independent biological replicates, the error bars indicate the standard deviation. Each biological replicate represents one data point. ns: non-significant. The datapoints are colored according to the *Pa* strain used.

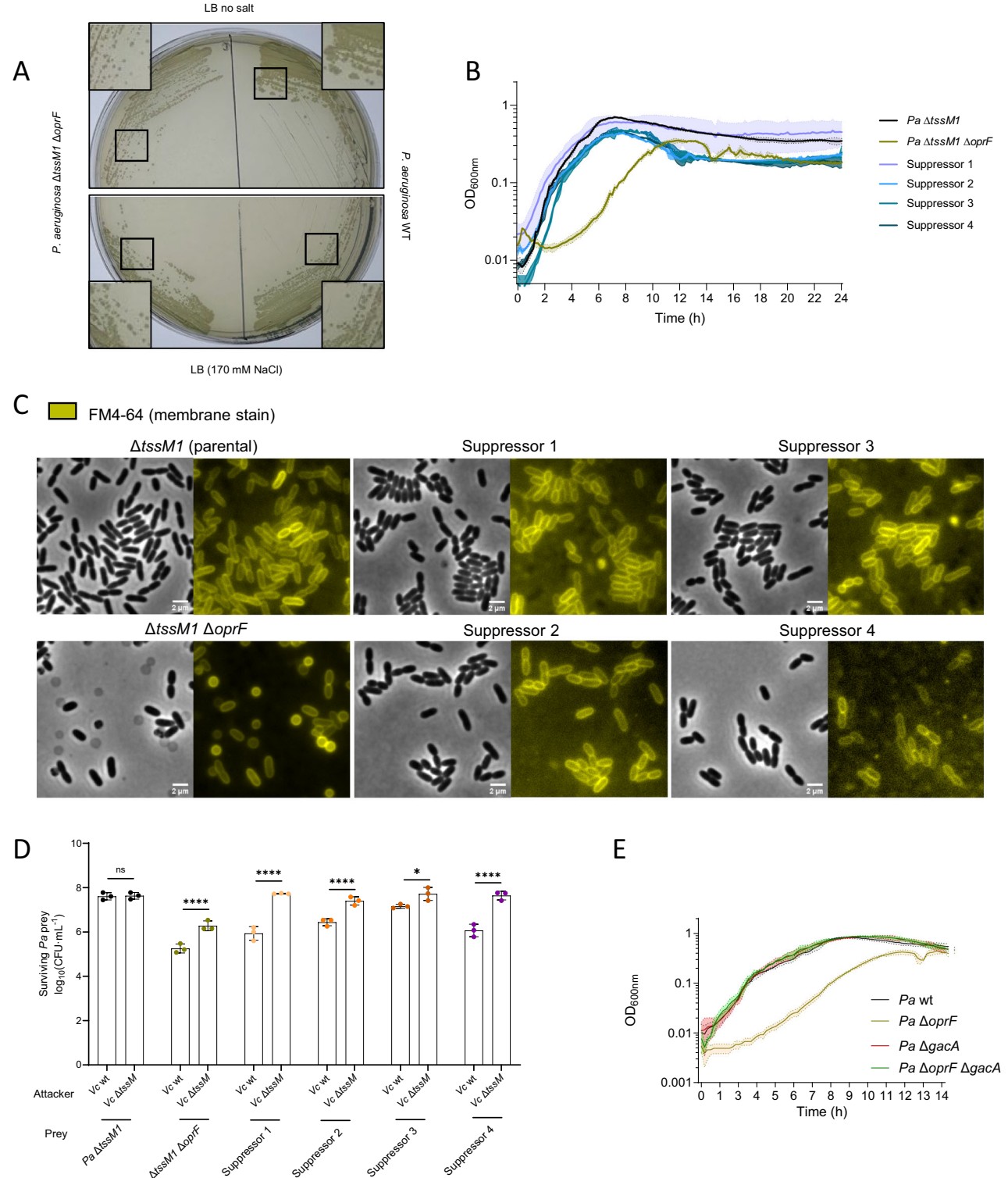

**Fig. 6 | ΔoprF suppressors grow normally and recover wild-type morphology but fail to resist T6SS attacks. A** Image of *P. aeruginosa* strains lacking *tssM1* and *oprF* or not, growing on an LB without salt or regular LB (170 mM NaCl) agar plate after overnight incubation at 37 °C. **B** Growth curves of parental and suppressor strains of *P. aeruginosa* (*Pa*) Δ*tssM1* Δ*oprF* in LB without salt. Mean OD$_{600nm}$ is indicated by continuous lines, and the shading around the lines represents the standard deviation. **C** Fluorescence microscopy images of parental and suppressor strains of *P. aeruginosa* lacking *tssM1* and *oprF*. In gray scale, phase contrast channel. In yellow, staining with FM4-64 (membrane dye). These are representative micrographs from three biological replicates. **D** CFU counts showing the survival of prey *P. aeruginosa* strains when competed against T6SS-active or –inactive *V.*

*cholerae* (*Vc*) in a 1:1 attacker to prey ratio. The bars indicate the mean across three independent biological replicates, the error bars indicate the standard deviation. Each biological replicate represents one data point. Data were analyzed with ordinary two-way ANOVA and Sidak's correction for multiple comparisons. Statistically significant adjusted p-values are indicated with asterisks and are *0.0177 and ****<0.0001. The datapoints are colored according to the *Pa* strain used. **E** Growth curve of wild-type and mutant *P. aeruginosa* strains lacking *oprF* and *gacA* or not in LB without salt. Mean OD$_{600nm}$ is indicated by continuous lines, and the shading around the lines represents the standard deviation. Source data are provided as a Source Data file.

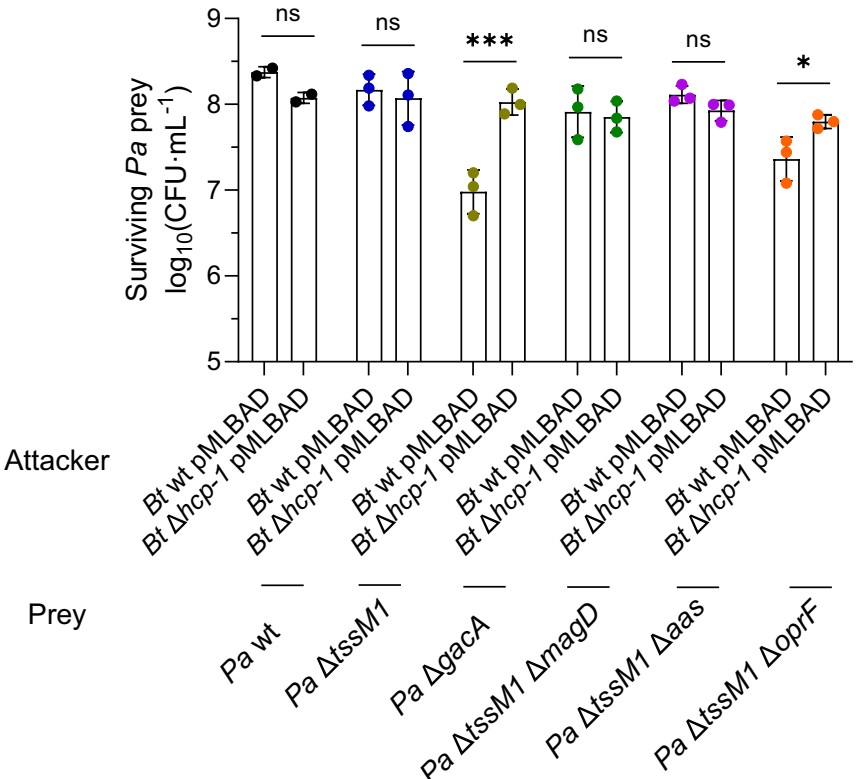

**Fig. 7 | T6SS resistance mechanisms are species- or effector-specific.** CFU counts showing the survival of prey *P. aeruginosa* (*Pa*) strains lacking different T6SS resistance mechanisms when competed with *B. thailandensis* (*Bt*) in a 10:1 attacker to prey ratio. Source data are provided as a Source Data file. The bars indicate the mean across three independent biological replicates, the error bars indicate the standard deviation. Each biological replicate represents one data point. Data were analyzed with multiple unpaired t tests. Statistically significant p-values are indicated with asterisks and are *0.0468 and ***0.0036.The datapoints are colored according to the *Pa* strain used.

whereas VgrG3 has conserved residues with the catalytic sites of lysozyme-like chitosanases[39], and therefore hydrolyzes the polysaccharidic chains of the PG[40,41]. The mechanism of Tae1 of *A. baylyi* has not been clearly elucidated, and it is unclear whether it is an amidase or a peptidase[15]. This raises the possibility MagD protects against PG-targeting effectors in a mechanism-dependent manner. Alternatively, the lack of a protective role against TseH might be explained by a low activity of TseH against *P. aeruginosa* under our experimental conditions, as the toxicity of TseH is highly dependent on the prey species (high activity against *Aeromonas* but no activity against immunity-deficient *V. cholerae* or *E. coli*[38]) and on abiotic factors such as the $Mg^{2+}$ and $Ca^{2+}$ concentration[42]. Interestingly, MagD was previously found to co-immunoprecipitate with TssM1[23], a structural transmembrane component of the H1-T6SS of *P. aeruginosa*, suggesting that these two components might co-localize where a foreign T6SS attack was detected and contribute to the defense against the injected effectors (in the case of MagD) and the assembly of a counterattack H1-T6SS machine (in the case of TssM1).

*P. aeruginosa* also possesses other resistance mechanisms that are not known to be regulated by GacA (Fig. 2D), such as the OMP OprF. OprF is the most abundant OMP in *P. aeruginosa*, and carries out multiple functions, including an outer membrane porin as well as anchoring the outer membrane to the PG[29]. Here, we found that OprF contributes to resistance to multiple T6SS effector classes (Fig. 5A, B). We confirmed this phenotype by complementation with a wild-type or mutant variant of OprF that no longer binds PG (Fig. 5C), and found that this variant is similarly able to promote survival against both *V. cholerae* 2740-80 and *A. baylyi* ADP1 suggesting that other functions of

OprF besides OM anchoring must be important for survival to T6SS attacks.

Our study also provides evidence for the genetic relationship between *oprF* and the *gacA/S* TCS. A strain lacking *oprF* displays a growth defect in LB that is more pronounced in LB without salt (Supplementary Fig. 4C). In LB without salt, this strain spontaneously reverts to wild-type-like growth at high frequency (Fig. 6A, B) and, although these suppressors also display wild-type-like morphology (Fig. 6C), they are still unable to survive T6SS attacks from *V. cholerae* 2740-80 (Fig. 6D). Our whole genome sequencing analysis showed that these suppressors carry a loss-of-function deletion of *gacS*, which explains why these strains are susceptible to T6SS attacks. Indeed, the GacA/S TCS is responsible for the growth defect of the Δ*oprF* strain, since deletion of *gacA* restores this defect (Fig. 6E). Further research is needed to determine why the expression of the GacA regulon leads to a growth defect in the Δ*oprF* genetic background.

Next, we also assessed whether our identified resistance mechanisms also play a role against *B. thailandensis* E264. As expected from a transposon screen performed by Ting et al.[18], *magD* does not play a role in defending from attacks from *B. thailandensis*, despite *B. thailandensis* encoding a PG-targeting amidase (Tae2)[43]. As discussed above, this may be due to differences in the mechanism of Tae2 compared to other PG-targeting effectors, or to low activity levels of this particular effector against *P. aeruginosa*. However, both GacA and OprF are important for defense against the T6SS of *B. thailandensis* (Fig. 7). Interestingly, the *arc2* operon identified by Ting et al.[18] did not appear in our screen. Taken together, our results show that *P. aeruginosa* possesses an array of T6SS resistance mechanisms, frequently

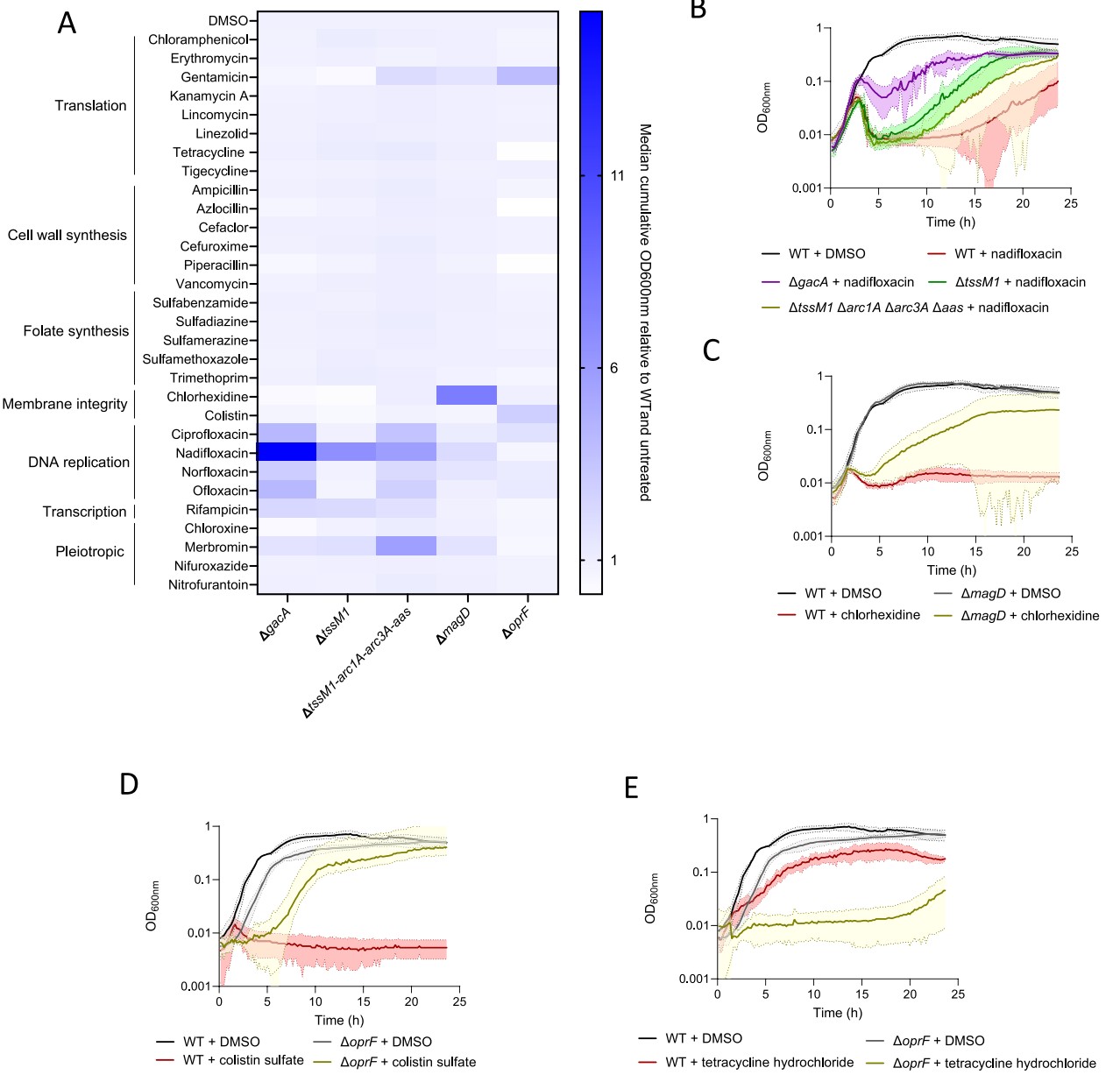

**Fig. 8 | T6SS resistance mechanisms influence antibiotic susceptibility. A** Heat-map showing the median cumulative OD$_{600\,nm}$ after 23.5 h of growth at 37 °C, relative to the wild-type and the untreated condition (DMSO control). **B** Growth curves of *P. aeruginosa* strains with or without 20 μM nadifloxacin, **C** 20 μM chlorhexidine, **D** colistin or **E** tetracycline. Mean OD$_{600nm}$ is indicated by continuous lines, and the shading around the lines represents the standard deviation. Source data are provided as a Source Data file.

under the control of the GacS/A TCS, that help it survive attacks from a variety of aggressor species that it can encounter in the environment. However, these resistance mechanisms might only protect against effectors with specific modes of action carried by specific species (Fig. 9).

Interestingly, other general stress response factors, such as those previously found to protect *E. coli* and *P. aeruginosa*[38,44], were not identified in our screen. In the case of *P. aeruginosa*, those factors were identified upon overexpression of TseL in the host periplasm, a condition that is likely more deleterious for *P. aeruginosa* than a single T6SS attack, where only a few molecules of TseL may be injected at the same time[44]. This would suggest that members of the GacA regulon mount a first layer of defense and that, when they are overwhelmed, general stress response mechanisms of the cell come into play. Performing a similar screen in a Δ*gacA* genetic background, where

dedicated protection mechanisms are not expressed, might uncover such general stress response mechanisms.

We also wondered whether these T6SS protection mechanisms could be involved in resistance to other antimicrobials, such as antibiotics. We screened a panel of 30 antibiotics against the wild-type, Δ*gacA*, Δ*tssM1*, Δ*tssM1* Δ*arc1A* Δ*arc3A*, Δ*aas*, Δ*magD* and Δ*oprF* and strains (Fig. 8 and Supplementary Figs. 6–8) and found that, surprisingly, strains lacking these T6SS resistance mechanisms display a slight growth advantage in the presence of selected antibiotics as compared to wild-type, suggesting that these T6SS protection mechanisms may worsen the toxicity of antibiotics. Of note, *tssM1*, *arc1A*, *arc3A*, *aas* and other members of the *gacA* regulon increase sensitivity to nadifloxacin and other fluoroquinolones. Part of this effect may be attributed to the ability of *arc3A* and *aas* to detoxify lysophospholipids that may be generated as a consequence of reactive oxygen species (ROS)[45,46] that are produced

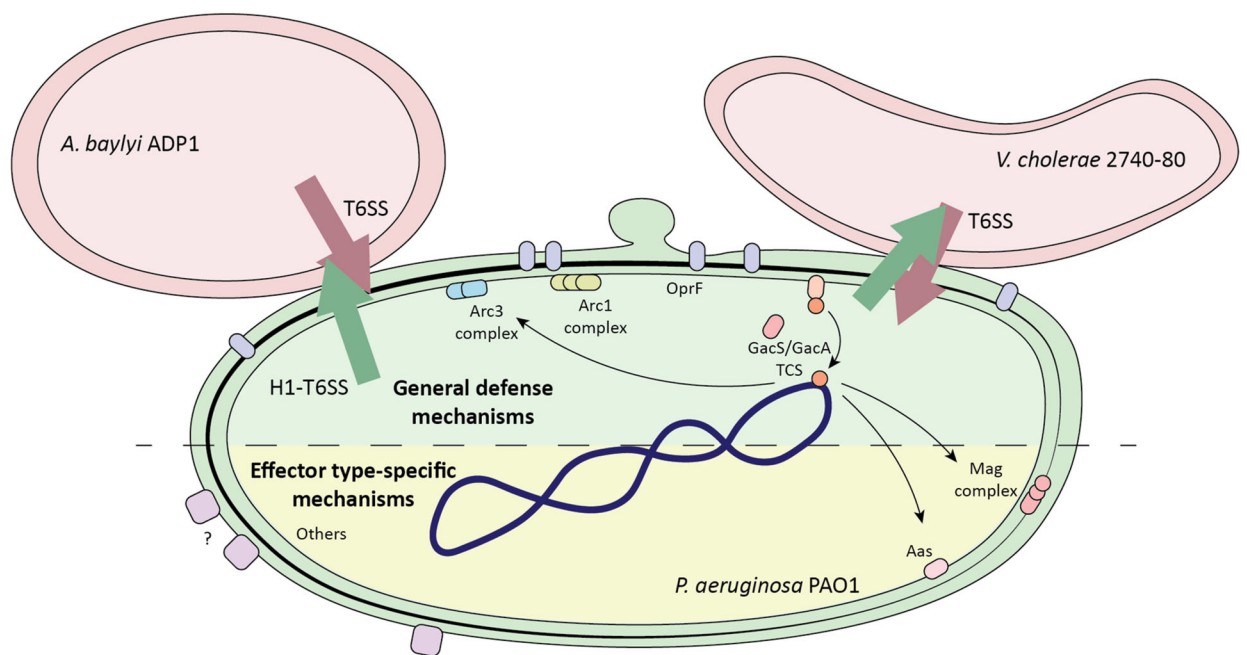

**Fig. 9 | _P. aeruginosa_ possesses multiple resistance mechanisms against T6SS attacks.** Model summarizing the different mechanisms identified in this study that contribute to the protection of _P. aeruginosa_ against T6SS attacks from _A. baylyi_ ADP1 and _V. cholerae_ 2740-80. These include mechanisms that are specific to certain effector types that may be carried by a subset of strains (like the Mag complex and Aas) and more general protection mechanisms that exert protection against the effectors carried by multiple attacker species (like the Arc1 and Arc3 complexes, the OprF OMP and the GacS/GacA TCS).

upon fluoroquinolone treatment[47], although further research is required to pinpoint the specific mechanism and other genes of the GacA regulon that may impact fluoroquinolone resistance. Interestingly, a Δ_magD_ mutant grows better in the presence of chlorhexidine, suggesting a further impact of MagD on the _P. aeruginosa_ envelope. As a consequence, we hypothesize that there are trade-offs to obtaining T6SS protection mechanisms that may lead to higher antibiotic susceptibility in _P. aeruginosa_. Similar trade-offs were also identified in _E. coli_ that evolved resistance against T6SS attacks from _V. cholerae_[48].

It is known that colistin kills gram-negative bacteria by targeting LPS and disrupting the OM[49,50]. Most mechanisms of resistance to colistin have been linked to LPS modification[49]. More research is needed to clarify why an _oprF_ mutant, that displays defects in OM stability, is more resistant to colistin. On the other hand, a Δ_oprF_ strain is also more susceptible to antibiotics, such as the ribosome-targeting tetracycline, or the beta-lactams piperacillin or azlocillin. These phenotypes cannot be explained by the porin function of _oprF_[29], since the strain should be more resistant to these antibiotics if _oprF_ was required for drug uptake. Overall, our findings serve as proof-of-concept that the study of the mechanisms of protection against the T6SS can lead to insights about antimicrobial resistance in bacteria.

In conclusion, in this work, we identified multiple protection mechanisms in _P. aeruginosa_ against T6SS attacks from _A. baylyi_ ADP1 and _V. cholerae_ 2740-80, which are also linked with resistance to classical antimicrobials. Interestingly, some T6SS resistance mechanisms may lead to increased sensitivity to antibiotics, suggesting that these mechanisms may not be generally beneficial. Further research will be needed to determine whether these protection mechanisms play a role in the survival of _P. aeruginosa_ strains in naturally occurring polymicrobial communities, and particularly in the context of clinical infections, and in the presence of antibiotics.

## Methods
### Bacterial strains and growth conditions
Bacterial strains used in this study can be found in Supplementary Data 2. Bacteria were grown in LB at 37 °C unless otherwise stated. _E._

_coli_ DH5α λpir was used as a cloning strain, and _E. coli_ SM10 λpir was used for the conjugation of _P. aeruginosa_ and _V. cholerae_. Antibiotics used were streptomycin (100 μg/mL for _V. cholerae_ and _A. baylyi_ on solid media; and 50 μg/mL for _A. baylyi_ in liquid media), gentamicin (20 μg/mL for _P. aeruginosa_ and _E. coli_), kanamycin (50 μg/mL for _A. baylyi_), carbenicillin (150 μg/mL for _E. coli_ and _V. cholerae_ and 100 μg/mL for _P. aeruginosa_), irgasan (20 μg/mL for _P. aeruginosa_) and trimethoprim (200 μg/mL for _B. thailandensis_ and 50 μg/mL for _E. coli_).

### Mutant construction
All plasmids used in this study are listed in Supplementary Table 1, and primers in Supplementary Data 3. _P. aeruginosa_ mutants were constructed as previously described using the pEXG2 suicide plasmid[51]. Briefly, 800 bp homology arms were amplified by PCR using primers (a list of all primers can be found in Supplementary Information 11) with 20 bp stretches that were homologous to the _Hind_III and _Xba_I restriction sites of the pEXG2 plasmid. PCR products were inserted into linearized pEGX2 plasmid by Gibson assembly[52] (NEB, E2611L). The resulting vectors were transformed into chemically-competent _E. coli_ DH5α λpir, selected on LB plates with 20 μg/mL gentamicin and verified by colony PCR and Sanger sequencing. The purified plasmids were then transformed into chemically competent _E. coli_ SM10 λpir, and the resulting strains were used for conjugation of _P. aeruginosa_ PAO1. Selection of _P. aeruginosa_ transconjugants was performed on LB with 20 μg/mL gentamicin and 20 μg/mL irgasan. The pEXG2 backbone was then curated by restreaking on LB plates without salt supplemented with 20 μg/mL irgasan and 10% sucrose, which were incubated for 36 h at room temperature. Colonies were screened by colony PCR with flanking primers.

pPSV35-derived plasmids were introduced into _P. aeruginosa_ by electroporation of 20–200 ng of plasmid into 1 mL of overnight stationary phase cultures washed twice with sterile water using a Bio-Rad GenePulser with the following settings: 25 μF, 400 Ω, 1.8 kV. All steps were performed at room temperature. 1 mL of LB medium was added immediately after the pulse and cells were incubated 1.5–2 h at 37 °C with shaking before plating on LB plates containing 20 μg/mL gentamicin.

The chromosomal insertion of *dCas9* in the *attnT7* site was performed as previously described[53]. Briefly, an 18–22 h old culture of *P. aeruginosa ΔtssM1* in LB without salt was washed twice with sterile water and resuspended in 100 μL sterile water. The bacterial suspension was electroporated with two plasmids at the same time, pUC18T: TNT7: dCas9Spas and pTNS2 that contains the Tn7 integrase and is unable to replicate in *P. aeruginosa*. The electroporation was performed as described above. The cells were recovered for 1 h in LB at 37 °C and 100 μL were plated on LB agar plates with 20 μg/mL gentamicin to select for transconjugants that integrated *dCas9* into the chromosome. The next day, colonies were grown in LB no salt and, as above, electroporated with plasmid pFLP2 containing the flippase that removes the backbone of the pUC18T plasmid. After recovery, 100 μL were plated on LB agar plates with 100 μg/mL carbenicillin and incubated overnight at 37 °C. The resulting colonies were restreaked on LB no salt + 10% sucrose to counterselect pFLP2. The successful insertion of *dCas9* was verified by PCR and the resulting strain was phenotypically validated by expression of an *ftsZ*-targeting sgRNA that led to cell filamentation as described[21].

*V. cholerae* mutants were constructed in a similar manner, using the pWM91 suicide plasmid and the *Xba*I and *Hind*III restriction sites. *V. cholerae* transconjugants were selected on LB plates with 150 μg/mL carbenicillin and 100 μg/mL streptomycin. The pWM91 backbone was curated by restreaking on LB plates without salt supplemented with 100 μg/mL streptomycin and 10% sucrose.

*A. baylyi* mutants were constructed by natural transformation as previously described[15], using a kanamycin resistance, streptomycin counterselectable, cassette and 800 bp homology arms.

*B. thailandensis* were conjugated with pMLBAD plasmid as previously described[54].

## Quantitative killing assays
Overnight cultures of *P. aeruginosa*, *V. cholerae*, *E. coli* and *B. thailandensis* were grown in LB at 37 °C and 200 rpm without antibiotics unless *P. aeruginosa* or *B. thailandensis* contained plasmids. *A. baylyi* overnight cultures were grown at 30 °C in similar conditions without antibiotics. Day cultures were launched in LB at the following dilutions: 1:100 for *P. aeruginosa*, 1:50 for *A. baylyi* and *B. thailandensis*, 1:200 for *E. coli* and 1:400 for *V. cholerae*. After 2.5–3 h, when all cultures reached an $OD_{600nm}$ of 0.7–1.4, cultures were spun down at 12,000 g for 2 min at room temperature. The cultures were concentrated to OD 10 with LB and 1:10 prey:attacker mixes were prepared. Five μL of the resulting mixes were spotted on LB agar plates, in technical triplicates, and allowed to dry. The plates were incubated at 37 °C for 3-4 h and then the spots were excised and resuspended in 500 μL liquid LB by vortexing. Ten-fold serial dilutions were performed and 5 μL of the resulting suspensions were spotted on selective plates. As selective agents, we used irgasan 20 μg/mL for *P. aeruginosa*, streptomycin 100 μg/mL for *V. cholerae* and *A. baylyi*, gentamicin 20 μg/mL for *E. coli* and 200 μg/mL trimethoprim for *B. thailandensis*. Plates were incubated overnight at 37 °C and the surviving CFUs were counted the next day and technical triplicates were averaged. The experiments were performed with at least three biological replicates and plotted with Prism.

## CRISPRi library electroporation
A *P. aeruginosa ΔtssM1* strain containing *dCas9* at the *attnT7* site was grown overnight in LB without salt and 4 mL were electroporated with 2 μL of the CRISPRi library as described above. Details on the library composition and experimental validation can be found in ref. 53. After electroporation, the bacterial suspension was resuspended in 1 mL LB and transferred to a 1 L Erlenmeyer flask containing 100 mL LB and allowed to recover for 1–1.5 h at 37 °C while shaking. Next, 20 μg/mL gentamicin was added and the culture was incubated overnight at 37 °C while shaking. This protocol led to an electroporation efficiency of $3.8 \cdot 10^6$ cells/mL, which allows for achieving a good coverage of the ca. 83,000 sgRNA library.

## CRISPRi screening competition
Three 100 mL day cultures in LB + 20 μg/mL gentamicin+1 mM IPTG were launched by inoculation of 1 mL at $OD_{600nm}$ 1 from the library electroporation culture. In parallel, 20 mL day cultures were launched from overnight cultures of *V. cholerae* and *A. baylyi* strains. The cultures were incubated for around 3 h until an $OD_{600nm}$ of 0.7–1.4 was reached. Five mL of each *P. aeruginosa* culture were taken, pelleted and frozen at −80 °C until further processing (time 0 before the competition started). For each replicate, the cultures were concentrated to OD 10 and mixed in a 10:1 *V. cholerae*:*P. aeruginosa* ratio or in a 1:1 *A.baylyi*:*P. aeruginosa* ratio and 90 μL of the resulting mixes were spotted on LB + 1 mM IPTG plates. *P. aeruginosa* alone was also spotted (no co-incubation control). The spots were allowed to dry and were then incubated at 37 °C for 3 h. Then, the spots were excised and resuspended in 7 mL LB by vortexing and pipetting. The resuspended bacteria were spun down for 10 min at 3,900 g and resuspended in 7 mL LB. Five mL were pelleted and frozen at −80 °C until further processing (samples after the first round of competition). The remaining 2 mL were used to measure the $OD_{600nm}$ and inoculate new day cultures in 100 mL LB + 20 μg/mL gentamicin + 1 mM IPTG with 1 mL OD 1. New day cultures of *V. cholerae* and *A. baylyi* were launched similarly to earlier. The same process was repeated for a second round of competition and the resulting samples were again pelleted and frozen at -80 °C until further processing.

## CRISPRi sequencing library preparation
Pellets were thawed and resuspended in 500 μL 50 mM NaOH. The samples were incubated for 10 min at 50 °C and 1 μL was used as template for a PCR reaction using Phusion DNA Polymerase (M0530S, NEB) as described[53] using primer pairs oATA402 + 403, oATA404 + 405, oATA406 + 407 and oATA408 + 409 for a fourth of the samples each. The PCR cycles were performed as follows: 98 °C for 3 min, 35 cycles of 98 °C denaturation for 10 s, 63 °C annealing for 10 s and 72 °C elongation for 10 s, followed by a final extension at 72 °C for 1 min and hold at 10 °C until further processing. The PCR products were run on a 2% agarose gel and purified from gel using a NucleoSpin Gel and PCR Clean-up kit (Macherey-Nagel). The DNA concentration of the samples was measured and samples were diluted to 1 ng/μL in 10 μL, of which 5 μL were used as template for the indexing PCR.

Indexing was performed with the kit IDT for Illumina Nextera DNA UD Index Set C (ref 20026934) with a PCR mix as follows: 5 μL DNA template, 5 μL primer mix, 5 μL Phusion GC Buffer, 0.5 μL 10 mM dNTPs, 0.25 μL Phusion DNA polymerase (M0530S, NEB) and 9.25 μL $H_2O$. The following program was used: 98 °C for 3 min, followed by 8 cycles of 98 °C denaturation for 10 s, 55 °C annealing for 10 s and 72 °C elongation for 10 s, and then a final extension at 72 °C for 1 min and hold at 10 °C until further processing. The PCR products were purified using 1.8 volumes of NEBNext Sample Purification Beads (E7767S, NEB). Samples were resuspended in 20 μL $H_2O$ and a quality check was run on a TapeStation 4150 device (Agilent) using DNA ScreenTape chips (Agilent) as recommended by the manufacturer. Sequencing was performed as previously described [54] by Illumina sequencing at the Genomics Facility Basel (ETH Zürich, University of Basel) using a NextSeq SR 81/10 kit.

Data analysis was performed using the European Galaxy server[55] using the previously described MAGeCK pipeline[56] using median normalization, FDR-adjusted threshold of 0.25, FDR as p-value adjustment method and 10 as remove zero threshold. The median method as used for the gene log-fold change. Data was plotted using Prism.

## Sample preparation for proteomics
Overnight cultures of *P. aeruginosa* were grown at 37 °C and 200 rpm, in biological triplicate. Day cultures in LB were launched (1:100 dilution of the overnight) and incubated for 2.5 3 h at 37 °C and 200 rpm, until they reached an $OD_{600nm}$ of 0.7–1.4. They were concentrated to an OD

of 10 and 20 µL were spotted on LB plates. The spots were allowed to dry and were then incubated at 37 °C for 3 h. The spots were then resuspended in 500 µL phosphate-buffered saline (PBS), pelleted by centrifugation at 12,000 g for 2 min and stored at -80 °C until further processing. Cell pellets were resuspended in lysis buffer (5% Sodium dodecyl sulfate (SDS), 10 mM tris(2-carboxyethyl) phosphine (TCEP), 100 mM Triethyloammonium bicarbonate (TEAB)) followed by incubation at 95 °C for 10 min. Cells were disrupted by ultrasonication using the PIXUL system (Active Motif) for 20 min with default settings (pulse 50 cycles, PRF 1 kHz, burst rate 20 Hz)) and the protein content was determined by tryptophan-based fluorescence assay (Infinite M Plex, Tecan). Sample alkylation was performed by addition of 20 mM iodoacetamide (IAA) and incubation at 25 °C for 30 min with gentle shaking. 10 µg of protein lysate was digested (trypsin 1/100, w/w; Promega) and purified using the SP3 protocol[57]. Peptide concentration was determined using an UV-based assay (Infinite M Nano, Tecan). Samples were then resuspended at a final concentration of 250 ng/µL.

### SWATH-MS data acquisition
Samples were separated on a Dionex UltiMate 3000 system (Thermo-Fisher Scientific) coupled online to an Orbitrap Exploris 480 mass spectrometer (ThermoFisher Scientific). In-house packed 20 cm, 75 µm ID capillary column with 1.9 µm Reprosil-Pur C18 beads (Dr. Maisch, Ammerbuch, Germany) was used. The column temperature was maintained at 60 °C using an integrated column oven interfaced online with the mass spectrometer. Formic acid (FA) 0.1% was used to buffer the pH in the two running buffers used. The total gradient time was 60 min and went from 2% to 12% acetonitrile (ACN) in 5 min, followed by 45 min to 35%, and 10 min at 50%. This was followed by a washout by 95% ACN, which was kept for 20 min, followed by re-equilibration of 0.1% FA buffer. Flow rate was kept at 300 nL/min. Spray voltage was set to 2500 V, funnel RF level at 40, and heated capillary at 275 °C. For DIA experiments full MS resolutions were set to 120,000 and full MS normalized AGC target was 300% with an IT of 45 ms. Mass range was set to 350–1400. Normalized AGC target value for fragment spectra was set at 1000%. In all, 63 windows of 9 Da were used with an overlap of 1 Da. Resolution was set to 15,000 and IT to 22 ms. Normalized CE was set at 28%. All data were acquired in centroid mode using positive polarity and advanced peak determination was set to on.

### SWATH-MS data analysis
For data processing and protein identification, raw data were imported into SpectroNaut (16.1.220730.53000, Biognosys) and analyzed with directDIA. Searches were carried out against a FASTA file including the proteomes of P. aeruginosa (UP000002438). Cysteine carbamidomethylation was set as fixed modification. Methionine oxidation, methionine excision at the N terminus were selected as variable modifications. Results were filtered for a 1% false discovery rate (FDR) on spectrum, peptide, and protein levels. Relative protein abundances and pairwise comparisons were calculated using the MSstats package[58] with default settings selected.

### CPRG lysis assay
For this assay, lacZ-deficient attacker strains were used, and P. aeruginosa prey strains carried an IPTG-inducible lacZ on the pME3856 plasmid. Overnight cultures of P. aeruginosa and V. cholerae were grown in LB at 37 °C and 200 rpm without antibiotics. A. baylyi overnight cultures were grown at 30 °C in similar conditions. Day cultures were launched in LB at the following dilutions: 1:100 for P. aeruginosa, 1:50 for A. baylyi and 1:400 for V. cholerae. After 2.5–3 h, when all cultures reached an OD$_{600nm}$ of 0.7–1.4, cultures were spun down at 12,000 g for 2 min at room temperature. The cultures were concentrated to OD 1 and mixed in a 1:2 prey:attacker ratio. Three µL of the resulting suspensions were spotted in wells of a flat-bottom 96-well plate that had been extemporaneously prepared with 150 µL LB with 1%

agar containing 0.1 mM IPTG (isopropyl-β-D-thiogalactoside) and 20 µg/mL CPRG (red-β-D-galactopyranoside) and pre-dried for 20 min at room temperature next to the Bunsen burner. The plates were incubated in an Epoch2 plate reader (BioTek) at 30 °C, and the OD$_{575nm}$ was measured every 10 min. The curves were plotted with Prism.

### Flow cytometry
Bacteria with fluorescently labeled T6SS sheath (TssB1-mNG) were grown to mid-exponential phase at 37 °C from an overnight culture in LB. Prior to measurement, the samples were diluted 100x in PBS and were analyzed in a BD Fortessa flow cytometer, with 950 V blue laser in low flow rate mode. Data were analyzed using FlowJo v10, and the median fluorescence intensity (MFI) was plotted using Prism.

### Growth curves
Overnight cultures of P. aeruginosa were diluted to OD$_{600nm}$ 1 in LB, and used to inoculate wells, in triplicate, of a flat-bottom 96-well plate containing LB or LB without NaCl, with or without antibiotics, at a starting OD$_{600nm}$ of 0.05. They were incubated at 37 °C in an Epoch2 plate reader (BioTek) with orbital shaking at 240 cpm and the OD$_{600nm}$ was measured every 10 min. If antibiotic treatment was performed, antibiotics were added after 1.5 h of incubation at 20 µM. The resulting curves were plotted in Prism.

### Fluorescence microscopy
Overnight cultures of P. aeruginosa were diluted 100-fold in LB and incubated for around 3 h at 37 °C and 200 rpm. They were then pelleted on a tabletop centrifuge at 12.000 g for 2 min at room temperature and resuspended to OD 10 with LB containing 4 µg/mL FM4-64 (T13320, ThermoFisher Scientific) and incubated for 10 min at room temperature in the dark. One µL of the cell suspensions were then placed on agar pads (1/3 LB, 2/3 PBS with 1% agarose) and imaged on a Nikon Ti-E inverted motorized microscope with Perfect Focus System and Plan Apo 100x Oil Ph3 DM (NA 1.4) objective lens. Spectra X light engine (Lumencor) and ET-mCherry (Chroma 49008) filter sets were used to excite and filter fluorescence and a sCMOS camera pco.edge 4.2 (PCO, Germany, pixel size 65 nm) and VisiView software (Visitron Systems, Germany) were used. Imaging was carried out at 30 °C by an Okolab T-unit (Okolab).

### Whole genome sequencing
Genomic DNA was extracted from overnight bacterial cultures in 1.5 mL LB following the instructions of the GenElute Bacterial Genomic DNA kit (NA2110, Sigma). DNA concentration was measured using the Qubit dsDNA Quantitation Broad Range Assay kit (Q32850, Thermo-Fisher Scientific) and barcoding was performed according to the manufacturer's instructions using the Rapid Barcoding Kit 24 V14 (SQK-RBK114.24, Oxford Nanopore Technologies). Samples were sequenced on a MinION flow cell (R10.4.1, FLO-MIN114, Oxford Nanopore Technologies). Basecalling was performed using Dorado software from Nanopore, and the genomes were compared using the Snakemake evo-genome-analysis pipeline that is publicly available at https://github.com/mmolari/evo-genome-analysis/. This pipeline uses the raw reads from each sample and maps it to the reference using Minimap2[59], extracting genomic changes including single nucleotide polymorphisms, gaps, insertions, clips and rearrangements.

### Statistics
When only two means were compared, we performed unpaired t-tests. When multiple means were compared, we used ordinary one-way analysis of variance (ANOVA) with Dunnett's multiple comparison test or two-way ANOVA with Sidak's multiple comparison test, using GraphPad Prism version 9.3.1. Unless otherwise stated, data are represented as mean ± standard deviation (SD). *p-value < 0.05, **p-value < 0.005, ***p-value < 0.0005, ****p-value < 0.00005, ns – p-value > 0.05.

**Reporting summary**

Further information on research design is available in the Nature Portfolio Reporting Summary linked to this article.

## Data availability

All mass spectrometry raw data files associated with this manuscript are accessible at MassIVE (https://massive.ucsd.edu) under accession number MSV000096798. Genome and CRISPRi screen sequencing data for this study have been deposited in the European Nucleotide Archive (ENA) at EMBL-EBI under accession number PRJEB84063. Source data are provided as a Source Data file and in the Zenodo repository with accession number 14811405 (https://zenodo.org/records/14811406). Source data are provided with this paper.

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

## Acknowledgements

This work was supported by the National Center of Competence in Research AntiResist funded by the Swiss National Science Foundation (51NF40_180541) to MB and UJ. We thank Mattia Zampieri for providing the antibiotic collection used in this study. We thank Marco Molari, Valentin Druelle and Leonardo Rocha for providing help with Nanopore sequencing sample preparation and data analysis. We thank the members of the Biozentrum FACS Core Facility (FCF) for help with flow cytometry and maintaining the Nanopore sequencing equipment. We thank the members of the Biozentrum Proteomics Core Facility (PCF) for help with mass-spectrometry data acquisition and analysis.

## Author contributions

Conceptualization: A.T.A. and M.B.; Methodology: A.T.A.; Software: A.T.A., M.A.V., and A.Kl.; Validation: A.T.A. and A.P.; Formal analysis: A.T.A. and M.A.V.; Investigation: A.T.A., A.P., and M.A.V.; Resources: A.K.a., A.K.l., U.J., and M.B.; Data curation: A.T.A.; Writing – original draft: A.T.A.; Writing – review & editing: A.T.A. and M.B.; Visualization: A.T.A. and A.P.; Supervision: A.T.A. and M.B.; Project administration: A.T.A. and M.B.; Funding acquisition: M.B. and U.J.

## Competing interests

The authors declare no competing interests.
