## [Transparent Peer Review file · Nature Communications]

Mechanisms of *Pseudomonas aeruginosa* resistance to Type VI Secretion System attacks

Corresponding Author: Dr Marek Basler

Version 0:

Reviewer comments:

Reviewer #1

(Remarks to the Author)

This study investigates how *Pseudomonas aeruginosa* defends itself against Type VI Secretion System (T6SS) attacks from competing bacterial strains. Using a combination of CRISPRi screening, genetic deletions, bacterial competition assays, and transcriptomic analysis, the authors identify key regulatory pathways involved in bacterial fitness and survival, including the GacA/GacS regulon and a GacA-independent factor, the outer membrane protein OprF. The findings contribute to our understanding of interbacterial antagonism and provide potential targets for antimicrobial strategies. While the study is easy to follow, several conclusions require further validation. The results are rich in phenotypic observations, but the mechanistic insights about how these regulatory pathways contribute to T6SS resistance are not fully explained.

Major issues:

1. In Lines 119–139, the authors propose that the Mag operon is critical for resistance against PG-targeting T6SS effectors, particularly demonstrating that MagD protects against VgrG3 from *V. cholerae* and offers partial protection against Tae1 from *A. baylyi*. However, the study also notes that TseH, another PG-targeting effector from *V. cholerae*, does not significantly impact the survival of the Δ magD mutant. The authors did not discuss why TseH fails to elicit the same effects as VgrG3 or Tae1, which raises questions about the specificity of MagD's protective role. Please explain whether TseH operates through a distinct mechanism or if MagD has a more specific affinity for certain PG-targeting effectors.
2. Among the various genes/mutations identified as influencing *P. aeruginosa*'s survival against T6SS attacks, only OprF was complemented, with restoration successfully rescuing T6SS resistance. However, other identified mutations were not complemented, making it difficult to confirm whether the observed phenotypes are directly due to these mutations or the result of secondary effects. Since genetic deletions can sometimes introduce compensatory changes or unintended regulatory shifts, complementation experiments should be conducted for additional key genes identified in the study.
3. Line 111, it was stated that Tse2 might play the most predominant role. A key control should be that tse2 deletion does not affect the overall T6SS secretion activities. If this has been shown, please include a reference.
4. Fig. 8A: The antibiotic resistance test was conducted using the " Δ tssM1-arc1A-arc3A-aas" strain. Would a single Δ tssM1 mutation alone affect antibiotic susceptibility? Including a Δ tssM1 single mutant as a control would help clarify its impact on the observed resistance phenotype.

Minor:

1. In the CPRG assay, did the authors use the Δ lacZ *Vibrio* mutant to rule out lacZ activity from lysed killer cells? Additionally, as wild-type *Pseudomonas* does not have lacZ, was a reporter plasmid or an engineered mutant strain used? This should be explicitly mentioned in the methods.
2. Fig. 1C: Some data points appear below the detection limit—could the authors clarify why this occurs?
3. Fig. 2B & 2C: Specify in the legend the killer:prey ratio used in the volcano plots.
4. In several figures, such as Fig. 3B and Fig. 4A, the color differences between lines are too subtle, making it difficult to distinguish between groups. Consider adjusting the color scheme to improve clarity.
5. FigS3A & S4A: certain killing assay groups lack survival data for *V. cholerae*.
6. FigS3D: The legend is missing.
7. Line172: figure citation should not include S4E.
8. FigS4C & S4D: in the figure, some instances of Pa are italicized while others are not.

Reviewer #2

(Remarks to the Author)

The manuscript by Tejada-Arranz et al. discusses the mechanisms of resistance that *Pseudomonas aeruginosa* has developed against attacks from the T6SS of competing bacteria. Specifically, it investigates how *P. aeruginosa* H1-T6SS responds to assaults from bacteria such as *Acinetobacter baylyi* and *Vibrio cholerae*. Using CRISPRi screening, the authors identify pathways contributing to T6SS resistance and suggest connections between these mechanisms and antibiotic susceptibility.

While the topic is relevant to the T6SS research community, this is not the first study to investigate *P. aeruginosa* resistance. The authors confirm previously identified factors, including the GacS/GacA two-component system and its regulated systems (Arc1-3 and aas), while also identifying two novel elements: the mag operon and the oprF gene. This discovery may be attributed to the use of different attacker cells compared to previous studies. Although the newly identified elements are interesting, the study lacks investigation into their molecular mechanisms.

Main points:

The authors are using a *P. aeruginosa* tssM1 to demonstrate that a strain without T6SS is still resistant to T6SS attack. However, *P. aeruginosa* has 3 systems, and although at different extent, all of them secrete antimicrobial effectors. In Fig. 1, when they see T6SS resistance to the attacker strains in a *P. aeruginosa* tssM1 mutant, they do not take into account that this strain can use the H2- and H3-T6SSs to attack the "attackers" (*Vibrio* and *Acinetobacter*). It would be interesting to see the results using a triple T6SS mutant.

Supplementary Figures 1A and 1B contain confusing labelling. The legend states "the survival of the attacker when competing against *A. baylyi* (A) and *V. cholerae* (B)" suggesting *P. aeruginosa* is the attacker, while the figure labels *A. baylyi* and *V. cholerae* as attackers. If the figures show *Acinetobacter* and *Vibrio* survival after competing with *P. aeruginosa*, it is puzzling that these strains are not killed by *P. aeruginosa*'s H1-T6SS. This outcome is expected for *A. baylyi* and *V. cholerae* tssM mutants since H1-T6SS would not counterattack, but is unexpected for wild-type strains.

The findings regarding Arc1-3 and aas merely confirm previous publications.

The gacA strain is more sensitive to T6SS attacks, as expected, because GacSA TC was previously described as involved in T6SS resistance in *P. aeruginosa* and confirmed in this study. However, the authors forgot to discuss that H2-T6SS is also regulated by this system (paragraphs 103-106) and could be another factor that has not been taken into account, despite showing this in Fig. 2D. Furthermore, Supplementary Figure 2 presents conflicting phenotypes for gacA and gacS mutants, with gacA being sensitive to *Vibrio* attack while gacS is not.

The authors proposed that the mag operon is involved in resistance to PG-targeting effectors. That seems to be clear for *V. cholerae* VgrG3 but not for Tae1 of *A. baylyi* (only one gene of the operon was a hit in Fig. 2C and almost no lysis differences between dtssM and dtssM dmagD in Fig. 3C). The authors should try other attacker strains with PG-targeting effectors to confirm this point. Curiously, the previous study by Ting et al 2022 did not identify this operon, although the attacker strain used in this study (*B. thal*) also has a PG-targeting effector (Tae2). This important point is not discussed in this study and contradicts the authors' statement that the mag resistance is effector-type dependent. It could be explained by different types of PG-targeting effectors, but since the authors do not attempt any molecular approach, there are no indications of this hypothesis. The CPRG-based lysis assay is poorly explained in the main text and the Figure legend, and it is difficult to interpret and distinguish the different colours.

The authors used two strains with 4 described effectors with a variety of functions, but they do not demonstrate that those are the only effectors in these strains. They should show that the deletion of the 4 effectors results in not T6SS killing. Moreover, they only show the resistance phenotype of individual effectors or the structural mutant. The resistance mechanism could be protecting from more than one type of effector at a time, especially for the OprF that the authors claim to be a general mechanism of resistance. The author should try deleting pair, triple and the 4 effectors to understand these mechanisms further. Again, the authors did not try to understand the OprF resistance mechanism at the molecular level, having only descriptive results. The authors indicate that the mechanism of OprF resistance might not be related to the PG binding function of this protein, but they have not proven that the mutant they are using to complement doesn't bind PG and only extrapolate results from *E. coli*.

The genetic link between gacS and oprF is, again, only observational, and there are no experiments to try to explain this relationship. Curiously, the 4 mutants they sequenced had the same 431-bp deletion in gacS, which might mean they are deleting an important domain for this link, or they have sequenced sister cells.

The authors propose that the immunity strategies are species-specific, which seems to contradict their other statement that the immunity is effector-type specific. How could this work? If a mechanism of resistance is protecting against PG-targeting effector, how might the origin (species) of the effector change the degree of resistance or sensitivity? If so, would the same effector delivered by different bacteria result in different resistance? It is difficult to come up with experiments to prove this hypothesis without having any idea of the mechanistic insight of these resistance mechanisms.

The final section, which explores connections between T6SS effector resistance and conventional antibiotic resistance, is comprehensive but purely descriptive. The authors make no attempt to elucidate the molecular mechanisms underlying their observations, many of which are unexpected. For example, if the mag operon is involved in PG-targeting antibiotics, why are these strains more sensitive to antibiotics targeting bacterial cell membranes? Similarly, why are strains sensitive to lipid-targeting T6SS effectors more resistant to DNA-targeting antibiotics like fluoroquinolones or nadifloxacin?

Minor correction:

Line 51: "multiple effectors that ARE highly active in vitro" (replacing "is" with "are")

Reviewer #3

(Remarks to the Author)

Major comments:

The ideas, experimental design, and overall significance of results that regard the screen and the identification of T6SS-effector resistance factors are convincing but they are delivered through a particularly challenging manuscript to read and understand. This presentation frustrates the most interested and experienced reader rather than enthrall and captivate. A plethora of figures are featured, each of which contributes a relatively small amount of information presented in a difficult-to-interpret manner (most comments below highlight this); this is not aided by accompanying text which is not sufficiently developed to tie all the details presented together seamlessly. In other words, the presentation is "scattered", hard to interpret and unfriendly to the reader. Perhaps the authors might consider finding another, more impactful approach or style to communicate and represent the results of the experiments by condensing figures, improve the narrative to be more informative, better contextualize experimental results, and contain a higher density of information per unit text. Finally, a graphic summary model that helps readers visualize a comprehensive and broad overview over all the details presented would be helpful as a "road map". The excessively parsimonious writing style in the manuscript (especially lack of explanations in the figure legends) does not do the results sufficient justice and the scattered choice of data representation makes it difficult to join the pieces into one, pleasant, coherent story. If the authors are not able to coalesce the text and figures articulately in one manuscript (admittedly, this is a challenge: lots of mutants, lots of assays), I suggest breaking down the data sets and publishing them separately in easier to deliver (and digest) "packages" (see comment below).

In the introduction the authors accurately state that bacteria have different types of T6SS effectors and immunity protein sets that are highly specific. However, even within their own species diversity in effector/immunity protein sets has been characterized. For example, within *V. cholerae* functionally divergent effectors were identified at each T6SS locus allowing members within the species to duel against one another with different outcomes (depending on the effector/immunity pairs). Strain 2740-80 therefore represents one "subset" of *V. cholerae* but not all members of this species. Of course, it is beyond the scope of this manuscript to examine all possible effector/immunity pairs competed against *P. aeruginosa*; however, the possibility of different outcomes given this effector/immunity diversity should, at least, be accounted for in the narrative and the choice of strains explained. For example, in each instance *V. cholerae* is mentioned, this should either be accompanied with the strain name or the broader group of strains harboring the same effector/immunity gene sets, as generalizations may not apply to all *V. cholerae* species. The same may apply to *A. baylyi* of which I am unaware such allelic nature of effector/immunity pairs has been shown, but could very well apply too. Either way, this notion is validated by the experiments the authors performed with *Burkholderia thailandensis*, which demonstrated a different resistance profile and should therefore be accounted for alternate effector/immunity pairs within the same species.

Special emphasis is dedicated to the contribution of *OprF*'s towards T6SS resistance only to be shown that an *oprF* deletion mutant has a conspicuous growth defect (in LB, exacerbated by the absence of NaCl). Complementation with two allelic forms of *oprF* restores both the growth defect as well as resistance against T6SS-mediated attack – as expected. Subsequently, *oprF* mutants plated on LB w/o NaCl yield CFUs which reveal a *gacS* deletion. As a result *gacS/oprF* deletion clones are constructed which restore the growth defect (whether this is in LB or LB w/o NaCl is not stated but should be) and retain T6SS sensitivity. This entire section of the manuscript describes phenotypes and features that, other than the relationship with *gacS* (which remains hereto unclarified) should be removed from this manuscript because there is no evidence that *oprF* specifically and directly mediates T6SS resistance/sensitivity phenotypes; the authors admit to this in lines 199-200 referring to this as a "generic resistance mechanism". This should be teased out in a follow-up because it is an interesting observation, but does not contribute much here because it remains an unresolved question.

The antibiotic screen experiments at the end come across as an observational afterthought, no explanation is offered, no mechanism tested.

Minor comments:

Lines 27 and 50: gram-negative ... pathogens/bacteria. Gram is only capitalized when referring directly to the stain, not when describing bacteria (ref: <https://wwwnc.cdc.gov/eid/page/preferred-usage>).

In the introduction (lines 50-51) the authors' state that the T6SS cluster of *A. baylyi* ADP1 and *V. cholerae* 2740-80 is highly active in vitro. This is accurate and the reason why this is the case is that these strains express the T6SS constitutively (without the requirement of external stimuli or induction), please add this here as it is more informative.

Albeit briefly, introduce *tssM* in the broader context of T6SS (not only as a structural component of *P. aeruginosa* H1-T6SS) since deletion mutants of this gene were used to disable T6SS in *A. baylyi* and *V. cholerae* alike.

Figure 1: Add the label "Surviving *Pa* Prey" to the Y-axis label of graphs A and B. Add "Surviving *E. coli* prey" to the Y-axis label of graph C. Please do the analogous to Supplementary Figures 1, this is particularly important in Supp. Fig 1 C where it is not intuitive that the attacking bacteria are being quantified on the Y-axis (this graph's labels are otherwise very similar to

Figure 1 C which can be confusing). Please do this to all applicable analogous graphs: 2E, 2F&G, 5B, 5C, 6D, 7, S2D-F, S3A, S4A-B, 4E-F, S5 as these additions will make the graphs understandable “at a glance”.

Regarding display of the same competitions but enumerating prey or attacker bacteria survival separately: have the authors considered exploring a mean to illustrate these same identical strains assayed against each other in the same graphs. This would save considerable space and condense some of these results coherently.

Figure 2: Could the authors place the words: *V. cholerae* 2740-80 and *A. baylyi* ADP1 over the volcano plots part A and B respectively? In the legend of the figure that matches these two plots, please add *P.aeruginosa* when referring to the “genes important for resistance to T6SS”.

Remove all instances of unnecessary use of “in order” throughout the manuscript (lines 80, 88, 194, 276) to improve readability and decrease word count. Along the same lines, it would be helpful to broaden the readership and interest of the article if authors could introduce the role, albeit briefly, of all genes and loci manipulated in experiments performed; for example, *retS*, *tssB1*, several others can be found throughout the manuscript that deserve improved descriptions.

In figure 2E indicate whether the difference in survival of *Pa tssM1* compared to *PA gacA* subjected to attach by *Ab wt* is statistically significant. Similarly, indicate where the differences shown in Figure 2D supplement are statistically significant.

The colored lines of the CPRG-based lysis assays are very thin and difficult to discern in the graph and the shading made this worse. I could not distinguish any of the strains plotted in graphs 3B-E, 4A and 4B. The figure legend does not explain the reason for the shadings; i.e. whether they have anything to do with error range or whatever maybe highlighted. Please indicate what OD 575nm is representative of along the Y axis of these analogous graphs for clarity. How many replicates of these experiments were conducted? Could these experimental results be presented differently? For example by calculating and graphing out the slopes; or perhaps in table format?

Please add sufficient details to all figure legends throughout the manuscript; they are far too brief and succinct and devoid of information to be helpful.

Figure 6A: Why are 3 colonies highlighted in the inset? This figure is missing WT struck on LB-NaCl, WT struck on LB no NaCl, and *oprF* struck on LB-NaCl for CFU size comparisons. Please add the missing controls (4 quadrants).

Figure 6B: The graph is un-interpretable: see previous comments about similar previous graphs, line thickness, colors, shading – none explained in the figure legend. Six strains appear to have been assayed; at least nine plots can be (barely) distinguished when zooming in.

Do the authors have an explanation for the large number of *oprF* clones grown on LB w/o NaCl of which 4 revealed to harbor 413 bp deletions within *gacS*? Do the authors expect that all several 100 CFUs would harbor the same deletion? Any speculations on the odds of this event and what it may mean? On the other hand, such considerations would be best reserved for a manuscript that clarifies the “genetic linkage” between *gacS* and *oprF*.

Why did the authors not pursue examination of the *gacS* strain antibiotic resistance and instead chose the *oprF* deletion strain which has a demonstrated growth defect?

Version 1:

Reviewer comments:

Reviewer #1

(Remarks to the Author)

The authors have adequately addressed all my concerns. The findings of this manuscript provide additional insights that advance our understanding of bacterial non-specific defense mechanisms against T6SS-mediated attacks.

Reviewer #2

(Remarks to the Author)

The authors have made an effort to address the comments that arose during the first round of revision. The manuscript has been improved, and I am happy with most of it. I only have a few additional very minor comments:

Line 31: H3-T6SS also delivers effectors involved in common goods.

Line 46: “composed of” instead of “composed by”

Line 84: misspelling in the word “cholerae”

Line 111-112: “compared” appears twice very closely. There are other examples with words like “important”, “resistance”, “such as” throughout the manuscript.

Line 112/196: parental strain. Line 247: antimicrobial compound (the nouns were missing)

Line 178: “or” instead of “nor”

The authors should be consistent with the word “wild-type”, sometimes it is written “wt”, others WT, on other occasions “wild

type” and lastly “wild-type”

Line 219: “without” instead “no”

Line 243 – space between “dtssM1” and “strain”

Line 464: bacteria “were” spun down

Line 468: similarly “to” earlier instead of “as”

Line 538: repetitive “plasmid-encoded” and “pME3856” plasmid

The structure “or not” should be substituted with “do not have” (lines 739, 742, 792, 800, 822, ...)

Line 752: genes “encoding proteins” important for ...

Figure legend 9: it is a very long sentence that could benefit from be split at least in 2.

REVIEWER COMMENTS

Reviewer #1 (Remarks to the Author):

This study investigates how *Pseudomonas aeruginosa* defends itself against Type VI Secretion System (T6SS) attacks from competing bacterial strains. Using a combination of CRISPRi screening, genetic deletions, bacterial competition assays, and transcriptomic analysis, the authors identify key regulatory pathways involved in bacterial fitness and survival, including the GacA/GacS regulon and a GacA-independent factor, the outer membrane protein OprF. The findings contribute to our understanding of interbacterial antagonism and provide potential targets for antimicrobial strategies. While the study is easy to follow, several conclusions require further validation. The results are rich in phenotypic observations, but the mechanistic insights about how these regulatory pathways contribute to T6SS resistance are not fully explained.

Major issues:

1. In Lines 119–139, the authors propose that the Mag operon is critical for resistance against PG-targeting T6SS effectors, particularly demonstrating that MagD protects against VgrG3 from *V. cholerae* and offers partial protection against Tae1 from *A. baylyi*. However, the study also notes that TseH, another PG-targeting effector from *V. cholerae*, does not significantly impact the survival of the Δ magD mutant. The authors did not discuss why TseH fails to elicit the same effects as VgrG3 or Tae1, which raises questions about the specificity of MagD's protective role. Please explain whether TseH operates through a distinct mechanism or if MagD has a more specific affinity for certain PG-targeting effectors.

We thank reviewer 1 for the comments on our manuscript. Indeed, previous structural studies suggest that TseH is likely a NlpC/P60 family cysteine endopeptidase¹, whereas VgrG3 has conserved residues with the catalytic sites of lysozyme-like chitosanases², and therefore hydrolyzes the polysaccharidic chains of the PG^{3,4}. This suggests that MagD may protect against lysozyme-like PG hydrolyses but not against endopeptidases. The mechanism of Tae1 of *A. baylyi* has not been clearly elucidated, and it is unclear whether it is an amidase or a peptidase⁵. Alternatively, the lack of a protective role against TseH might be explained by a low activity of TseH against *P. aeruginosa* under our experimental conditions. The toxicity of TseH was found to be highly dependent on the prey species, with high activity against *Aeromonas* but not against immunity-deficient *V. cholerae* or *E. coli*¹. Furthermore, its toxicity was shown to be dependent on abiotic factors such as the concentration of Mg²⁺ and Ca²⁺, suggesting that TseH might simply not be very active under our experimental conditions⁶. This was added to the discussion (lines 291-300).

2. Among the various genes/mutations identified as influencing *P. aeruginosa*'s survival against T6SS attacks, only OprF was complemented, with restoration successfully rescuing T6SS resistance. However, other identified mutations were not complemented, making it difficult to confirm whether the observed phenotypes are directly due to these mutations or the result of secondary effects. Since genetic deletions can sometimes introduce compensatory changes or unintended regulatory shifts, complementation experiments should be conducted for additional key genes identified in the study.

We complemented the other major gene that we identified in this study, MagD, in the native locus and under the control of the native promoter. Complementation results are shown in the updated figure 3A. The text has been adjusted accordingly (lines 143-144).

3. Line 111, it was stated that Tse2 might play the most predominant role. A key control should be that tse2 deletion does not affect the overall T6SS secretion activities. If this has been shown, please include a reference.

It was in fact previously shown by our laboratory that deletion of any or all of the effectors of *A. baylyi* ADP1, including Tse2, does not impact T6SS assembly and firing⁵. This reference is on line 122.

4. Fig. 8A: The antibiotic resistance test was conducted using the “ Δ tssM1-arc1A-arc3A-aas” strain. Would a single Δ tssM1 mutation alone affect antibiotic susceptibility? Including a Δ tssM1 single mutant as a control would help clarify its impact on the observed resistance phenotype.

The reviewer is correct, a Δ tssM1 control should be included. We did this experiment, together with a Δ gacA control as suggested by reviewer 3 and updated figure 8A accordingly. Surprisingly, deletion of tssM1 already impacts resistance to the fluoroquinolone nadifloxacin, leading to increased growth in the presence of the antibiotic compared to WT. However, deletion of arc1A, arc3A and aas in a Δ tssM1 background extends this effect to other fluoroquinolones such as ciprofloxacin, norfloxacin and ofloxacin. These fluoroquinolone-related effects were more pronounced in a Δ gacA strain, as one would expect since this strain does not express any of these T6SS-resistance mechanisms and potentially other genes that may be relevant for this phenotype. This information was added to the text on lines 235-242.

Minor:

1. In the CPRG assay, did the authors use the Δ lacZ Vibrio mutant to rule out lacZ activity from lysed killer cells? Additionally, as wild-type Pseudomonas does not have lacZ, was a reporter plasmid or an engineered mutant strain used? This should be explicitly mentioned in the methods.

Yes, we used a Vibrio mutant lacking lacZ, this was clarified in the material and methods section. For Pseudomonas, a reporter plasmid pME3856 was used, containing lacZ under the control of a P_{tac} promoter. This information was added to the materials and methods section (lines 532-533).

2. Fig. 1C: Some data points appear below the detection limit—could the authors clarify why this occurs?

Because they are under the detection limit, i.e. no colonies were detected. This was moved to the detection limit. Statistical significance was not affected by this change.

3. Fig. 2B & 2C: Specify in the legend the killer:prey ratio used in the volcano plots.

This information was added to the legend of Figure 2 (lines 745 and 747).

4. In several figures, such as Fig. 3B and Fig. 4A, the color differences between lines are too subtle, making it difficult to distinguish between groups. Consider adjusting the color scheme to improve clarity.

As recommended also by reviewer 3, we decided to change the representation of the results from CPRG lysis assays to improve clarity. Instead of showing lysis curves, we are now showing bar graphs with cumulative optical density at 575 nm after 10h. This highlights the differences between the

different curves and makes results more easily interpretable (currently Figures 3C-D and 4). The curves were moved to the supplementary materials (currently Supplementary Figures 3C—D and 3I-J).

5. FigS3A & S4A: certain killing assay groups lack survival data for *V. cholerae*.

These competitions were not relevant to the study and were not performed. The graphs have been cleaned up to remove the extra labels from the axis (currently Supplementary Figures 3B and 4A).

6. FigS3D: The legend is missing.

The legend of this figure (corresponding to Supplementary Figure 3I in the current version) was added.

7. Line172: figure citation should not include S4E.

Supplementary Figure 4F contains a necessary control for Figure 5C (survival of the attacker strains).

8. FigS4C & S4D: in the figure, some instances of Pa are italicized while others are not.

This was corrected, all instances are italicized now.

Reviewer #2 (Remarks to the Author):

The manuscript by Tejada-Arranz et al. discusses the mechanisms of resistance that *Pseudomonas aeruginosa* has developed against attacks from the T6SS of competing bacteria. Specifically, it investigates how *P. aeruginosa* H1-T6SS responds to assaults from bacteria such as *Acinetobacter baylyi* and *Vibrio cholerae*. Using CRISPRi screening, the authors identify pathways contributing to T6SS resistance and suggest connections between these mechanisms and antibiotic susceptibility. While the topic is relevant to the T6SS research community, this is not the first study to investigate *P. aeruginosa* resistance. The authors confirm previously identified factors, including the GacS/GacA two-component system and its regulated systems (Arc1-3 and aas), while also identifying two novel elements: the mag operon and the oprF gene. This discovery may be attributed to the use of different attacker cells compared to previous studies. Although the newly identified elements are interesting, the study lacks investigation into their molecular mechanisms.

Main points:

The authors are using a *P. aeruginosa* tssM1 to demonstrate that a strain without T6SS is still resistant to T6SS attack. However, *P. aeruginosa* has 3 systems, and although at different extent, all of them secrete antimicrobial effectors. In Fig. 1, when they see T6SS resistance to the attacker strains in a *P. aeruginosa* tssM1 mutant, they do not take into account that this strain can use the H2- and H3-T6SSs to attack the “attackers” (*Vibrio* and *Acinetobacter*). It would be interesting to see the results using a triple T6SS mutant.

We thank the reviewer for these comments. To date, the H2- and H3-T6SSs of *P. aeruginosa* have not been reported to retaliate to incoming attacks from other bacteria, and they lack the regulatory cascade responsible for damage sensing. Furthermore, under our experimental conditions and with our *P. aeruginosa* PAO1 strain, the H3-T6SS was found to be silenced⁷. Nevertheless, we performed the requested experiment, now replacing Figure 1A-B. As expected, the triple knockout behaves like a Δ tssM1 strain, indicating that there is no role of the H2- and H3-T6SSs in the observed phenotypes. The text has been adjusted accordingly (lines 78-79).

Supplementary Figures 1A and 1B contain confusing labelling. The legend states "the survival of the attacker when competing against *A. baylyi* (A) and *V. cholerae* (B)" suggesting *P. aeruginosa* is the attacker, while the figure labels *A. baylyi* and *V. cholerae* as attackers. If the figures show *Acinetobacter* and *Vibrio* survival after competing with *P. aeruginosa*, it is puzzling that these strains are not killed by *P. aeruginosa*'s H1-T6SS. This outcome is expected for *A. baylyi* and *V. cholerae* *tssM* mutants since H1-T6SS would not counterattack, but is unexpected for wild-type strains.

The sentence in the figure legend has been simplified for clarity. In this case, we refer to either *A. baylyi* or *V. cholerae* as the attacker strain, which outnumbered the prey (*P. aeruginosa*) in a 10:1 attacker:prey ratio. Under such conditions where the attacker vastly outnumbered the prey, we in fact expect that little to no killing of the attacker strain is observed, since whatever killing of the attacker takes place is compensated for by cell division during the competition time. However, if we alter the attacker:prey ratio, we can indeed observe killing of both *A. baylyi* and *V. cholerae* by *P. aeruginosa*, as shown in revised Supplementary Figures 1C-1F, both in a 1:1 ratio and when *P. aeruginosa* outnumbered the other species in a 10:1 ratio. This information has also been added to the text (lines 79-84).

The findings regarding *Arc1-3* and *aas* merely confirm previous publications.

Indeed, our findings regarding *arc3* merely reproduce what was already shown in the literature by Ting *et al.*⁸. However, the role for *aas* was merely hypothesized without phenotypic evidence that is provided by the present study. Lastly, Ting *et al.* could not pinpoint the effector class that *arc1A* protects against, while our study indicates that *arc1A* also protects against lipases. This was indicated in the study and has been further highlighted in the revised version (lines 178-180).

The *gacA* strain is more sensitive to T6SS attacks, as expected, because GacSA TC was previously described as involved in T6SS resistance in *P. aeruginosa* and confirmed in this study. However, the authors forgot to discuss that H2-T6SS is also regulated by this system (paragraphs 103-106) and could be another factor that has not been taken into account, despite showing this in Fig. 2D.

Indeed, the H2-T6SS is also regulated by GacA. However, the H2-T6SS does not play a role for resistance against T6SS under our experimental conditions, as a triple H1-, H2- and H3-T6SS knockout behaves like a $\Delta tssM1$ strain (updated Figure 1A-B).

Furthermore, Supplementary Figure 2 presents conflicting phenotypes for *gacA* and *gacS* mutants, with *gacA* being sensitive to *Vibrio* attack while *gacS* is not.

Some sgRNAs targeting *gacS* were also identified in our *A. baylyi* dataset as potential hits (updated Supplementary Figure 2B). However, overall *gacS* fell under the significance threshold. The hit is significant though in the case of the *V. cholerae* dataset. Given the well-characterized functional relationship between GacA and GacS, we did not further explore this minor discrepancy and attribute it to differential efficiency of the sgRNAs in the library, which may target *gacS* less efficiently than *gacA*, thus yielding weaker phenotypes. The dot corresponding to *gacS* was highlighted in Figure 2C.

The authors proposed that the *mag* operon is involved in resistance to PG-targeting effectors. That seems to be clear for *V. cholerae* VgrG3 but not for Tae1 of *A. baylyi* (only one gene of the operon was a hit in Fig. 2C and almost no lysis differences between *dtssM* and *dtssM* $\Delta magD$ in Fig. 3C). The authors should try other attacker strains with PG-targeting effectors to confirm this point. Curiously, the

previous study by Ting et al 2022 did not identify this operon, although the attacker strain used in this study (B. thai) also has a PG-targeting effector (Tae2). This important point is not discussed in this study and contradicts the authors' statement that the mag resistance is effector-type dependent. It could be explained by different types of PG-targeting effectors, but since the authors do not attempt any molecular approach, there are no indications of this hypothesis. The CPRG-based lysis assay is poorly explained in the main text and the Figure legend, and it is difficult to interpret and distinguish the different colours.

Indeed, a possible explanation for the specificity of MagD would be the molecular mechanism of the different PG-targeting effectors, plus the fact that the effectors may not be active in all possible preys (e.g. because of missing molecular target or different concentrations of potential cofactors). This has been included in the discussion (lines 292-300). It is beyond the scope of this study to research the specific mechanism of action of Tae2 and other PG-targeting effectors, but we have added these valid points to the discussion (lines 326-329). We have modified the graphs of the lysis assays to improve clarity (Figures 3C-D and Figure 4), displaying cumulative OD_{575nm} instead of the lysis curves. The lysis curves have been moved to the supplementary materials (Supplementary Figures 3C-D and 3I-J). The explanation of the CPRG assay was also improved (lines 146-148).

The authors used two strains with 4 described effectors with a variety of functions, but they do not demonstrate that those are the only effectors in these strains. They should show that the deletion of the 4 effectors results in not T6SS killing.

We performed the requested killing assays and show no killing of susceptible *E. coli* in the absence of the 4 known effectors of *A. baylyi* or upon inactivation of the 4 known effectors of *V. cholerae* 2740-80, suggesting that there are no more T6SS effectors that are active against this *E. coli* strain under our experimental conditions. The new data has been added to Fig. 1C and the text has been modified accordingly (lines 88-90).

Moreover, they only show the resistance phenotype of individual effectors or the structural mutant. The resistance mechanism could be protecting from more than one type of effector at a time, especially for the OprF that the authors claim to be a general mechanism of resistance. The author should try deleting pair, triple and the 4 effectors to understand these mechanisms further. Again, the authors did not try to understand the OprF resistance mechanism at the molecular level, having only descriptive results.

Upon multiple repetitions of this experiment with all multiple-effector inactivated strains that we have available, it became apparent that VgrG3 is the main driver of killing of the $\Delta oprF$ strain by *V. cholerae*, as all mutants lacking an active VgrG3 displayed a significant killing defect comparable to the T6SS-deficient strain and the strain carrying all 4 inactivated effectors. Figure 5A has been replaced with the updated panel. However, an *A. baylyi* strain lacking the corresponding PG-targeting effector, Tae1, was still able to kill *P. aeruginosa* $\Delta oprF$, indicating that other effector types are also able to kill this strain more than *P. aeruginosa* $\Delta tssM1$. This information has been updated accordingly in the text (lines 191-196).

The authors indicate that the mechanism of OprF resistance might not be related to the PG binding function of this protein, but they have not proven that the mutant they are using to complement doesn't bind PG and only extrapolate results from *E. coli*.

The *P. aeruginosa* strain lacking *oprF* displays an increased amount of round cells and cell ghosts derived from membrane blebbing or cell lysis. Although the expression of the WT allele of OprF reduces the amount of cell ghosts, expression of the R296E variant does not, suggesting that this mutant still displays morphological defects consistent with envelope stress and instability (new Supplementary Figure 4E).

The genetic link between *gacS* and *oprF* is, again, only observational, and there are no experiments to try to explain this relationship. Curiously, the 4 mutants they sequenced had the same 431-bp deletion in *gacS*, which might mean they are deleting an important domain for this link, or they have sequenced sister cells.

We agree with the reviewer. Although it is outside of the scope of this study to elucidate the genetic relationship between *gacS* and *oprF*, we found that the fact that the growth defect of a Δ *oprF* strain is compensated by inactivation of this two-component system is an interesting observation.

The authors propose that the immunity strategies are species-specific, which seems to contradict their other statement that the immunity is effector-type specific. How could this work? If a mechanism of resistance is protecting against PG-targeting effector, how might the origin (species) of the effector change the degree of resistance or sensitivity? If so, would the same effector delivered by different bacteria result in different resistance? It is difficult to come up with experiments to prove this hypothesis without having any idea of the mechanistic insight of these resistance mechanisms.

We understand the confusion generated by these statements. We believe that the immunity strategies are effector-type specific (including the particular mechanism of action of each effector type), and since different strains contain different effector sets of different types, the immunity is therefore also species-specific. This has been clarified in the text (lines 220 and 229-230; lines 334-335). Indeed, it would be interesting to test whether the same effector delivered by a different species maintains the same level of toxicity. However, given the poor current understanding of the structural signals mediating delivery of T6SS effectors, testing this is outside the scope of this study.

The final section, which explores connections between T6SS effector resistance and conventional antibiotic resistance, is comprehensive but purely descriptive. The authors make no attempt to elucidate the molecular mechanisms underlying their observations, many of which are unexpected. For example, if the *mag* operon is involved in PG-targeting antibiotics, why are these strains more sensitive to antibiotics targeting bacterial cell membranes? Similarly, why are strains sensitive to lipid-targeting T6SS effectors more resistant to DNA-targeting antibiotics like fluoroquinolones or nadifloxacin?

We expanded the discussion section and put forward a hypothesis why some members of the GacA regulon may be involved in resistance to fluoroquinolones (lines 351-357). However, further research is needed into the specific mechanism of action of MagD before we can speculate about its role on growth in the presence of chlorhexidine.

Minor correction:

Line 51: "multiple effectors that ARE highly active in vitro" (replacing "is" with "are")

This has been corrected.

Reviewer #3 (Remarks to the Author):

Major comments:

The ideas, experimental design, and overall significance of results that regard the screen and the identification of T6SS-effector resistance factors are convincing but they are delivered through a particularly challenging manuscript to read and understand. This presentation frustrates the most interested and experienced reader rather than enthrall and captivate. A plethora of figures are featured, each of which contributes a relatively small amount of information presented in a difficult-to-interpret manner (most comments below highlight this); this is not aided by accompanying text which is not sufficiently developed to tie all the details presented together seamlessly. In other words, the presentation is “scattered”, hard to interpret and unfriendly to the reader. Perhaps the authors might consider finding another, more impactful approach or style to communicate and represent the results of the experiments by condensing figures, improve the narrative to be more informative, better contextualize experimental results, and contain a higher density of information per unit text. Finally, a graphic summary model that helps readers visualize a comprehensive and broad overview over all the details presented would be helpful as a “road map”. The excessively parsimonious writing style in the manuscript (especially lack of explanations in the figure legends) does not do the results sufficient justice and the scattered choice of data representation makes it difficult to join the pieces into one, pleasant, coherent story. If the authors are not able of coalesce the text and figures articulately in one manuscript (admittedly, this is a challenge: lots of mutants, lot’s of assays), I suggest breaking down the data sets and publishing them separately in easier to deliver (and digest) “packages” (see comment below).

We thank reviewer 3 for his comments. We simplified some of the graphic representations of our assays, namely the CPRG lysis assays, and we now display the results as accumulated OD_{575nm}, making them more easily interpretable. We also added a graphic summary in Figure 9. We made edits for clarity throughout the text.

In the introduction the authors accurately state that bacteria have different types of T6SS effectors and immunity protein sets that are highly specific. However, even within their own species diversity in effector/immunity protein sets has been characterized. For example, within *V. cholerae* functionally divergent effectors were identified at each T6SS locus allowing members within the species to duel against one another with different outcomes (depending on the effector/immunity pairs). Strain 2740-80 therefore represents one “subset” of *V. cholerae* but not all members of this species. Of course, it is beyond the scope of this manuscript to examine all possible effector/immunity pairs competed against *P. aeruginosa*; however, the possibility of different outcomes given this effector/immunity diversity should, at least, be accounted for in the narrative and the choice of strains explained. For example, in each instance *V. cholerae* is mentioned, this should either be accompanied with the strain name or the broader group of strains harboring the same effector/immunity gene sets, as generalizations may not apply to all *V. cholerae* species. The same may apply to *A. baylyi* of which I am unaware such allelic nature of effector/immunity pairs has been shown, but could very well apply too. Either way, this notion is validated by the experiments the authors performed with *Burholderia thailandensis*, which demonstrated a different resistance profile and should therefore be accounted for alternate effector/immunity pairs within the same species.

Indeed, *V. cholerae* 2740-80 is a representative strain for environmental, non-toxigenic, AAA effector type strains. This information has been added to the introduction (lines 56-57) and the strain name

has been specified throughout the manuscript to avoid generalizations. To the best of our knowledge, strain diversity with regard to effector types has not been studied for *A. baylyi*, but the strain name ADP1 has also been specified throughout the manuscript to avoid a similar issue.

Special emphasis is dedicated to the contribution of OprF's towards T6SS resistance only to be shown that an *oprF* deletion mutant has a conspicuous growth defect (in LB, exacerbated by the absence of NaCl). Complementation with two allelic forms of *oprF* restores both the growth defect as well as resistance against T6SS-mediated attack – as expected. Subsequently, *oprF* mutants plated on LB w/o NaCl yield CFUs which reveal a *gacS* deletion. As a result *gacS/oprF* deletion clones are constructed which restore the growth defect (whether this is in LB or LB w/o NaCl is not stated but should be) and retain T6SS sensitivity. This entire section of the manuscript describes phenotypes and features that, other than the relationship with *gacS* (which remains hereto unclarified) should be removed from this manuscript because there is no evidence that *oprF* specifically and directly mediates T6SS resistance/sensitivity phenotypes; the authors admit to this in lines 199-200 referring to this as a “generic resistance mechanism”. This should be teased out in a follow-up because it is an interesting observation, but does not contribute much here because it remains an unresolved question.

We showed that the double *oprF gacA* deletion restores growth in LB no salt where the phenotype is strongest. This has been specified in the manuscript (line 216). However, OprF does mediate T6SS resistance, as shown by deletion and complementation of this gene. It is not clear whether this effect is due to one of the (many) structural functions of OprF or something else. It may be indirect, as the reviewer correctly identified, but we still believe that it confers resistance to multiple T6SS effector classes.

The antibiotic screen experiments at the end come across as an observational afterthought, no explanation is offered, no mechanism tested.

We expanded the discussion and suggested hypotheses about some of the mechanisms that may contribute to these observations (lines 351-357). Indeed, further research is needed to elucidate the mechanisms taking place here, and in particular the increased resistance to fluoroquinolones of multiple T6SS-susceptible strains.

Minor comments:

Lines 27 and 50: gram-negative ... pathogens/bacteria. Gram is only capitalized when referring directly to the stain, not when describing bacteria (ref: <https://wwwnc.cdc.gov/eid/page/preferred-usage>).

This error has been corrected throughout the manuscript.

In the introduction (lines 50-51) the authors' state that the T6SS cluster of *A. baylyi* ADP1 and *V. cholerae* 2740-80 is highly active in vitro. This is accurate and the reason why this is the case is that these strains express the T6SS constitutively (without the requirement of external stimuli or induction), please add this here as it is more informative.

This has been added (lines 53-54).

Albeit briefly, introduce *tssM* in the broader context of T6SS (not only as a structural component of P.

aeruginosa H1-T6SS) since deletion mutants of this gene were used to disable T6SS in *A. baylyi* and *V. cholerae* alike.

This has been added to the introduction (lines 36-38).

Figure 1: Add the label “Surviving Pa Prey” to the Y-axis label of graphs A and B. Add “Surviving *E. coli* prey” to the Y-axis label of graph C. Please do the analogous to Supplementary Figures 1, this is particularly important in Supp. Fig 1 C where it is not intuitive that the attacking bacteria are being quantified on the Y-axis (this graph’s labels are otherwise very similar to Figure 1 C which can be confusing). Please do this to all applicable analogous graphs: 2E, 2F&G, 5B, 5C, 6D, 7, S2D-F, S3A, S4A-B, 4E-F, S5 as these additions will make the graphs understandable “at a glance”.

We thank the reviewer for this comment, this change has been made throughout all figures.

Regarding display of the same competitions but enumerating prey or attacker bacteria survival separately: have the authors considered exploring a mean to illustrate these same identical strains assayed against each other in the same graphs. This would save considerable space and condense some of these results coherently.

We prefer to display survival of the attacker separately, since this is just a control that approximately the same number of attacker bacteria were present, and does not add anything further to the narrative while doubling the amount of bars to look at in any given graph. We feel that the representation is clearer like this.

Figure 2: Could the authors place the words: *V. cholerae* 2740-80 and *A. baylyi* ADP1 over the volcano plots part A and B respectively? In the legend of the figure that matches these two plots, please add *P.aeruginosa* when referring to the “genes important for resistance to T6SS”.

These changes have been incorporated.

Remove all instances of unnecessary use of “in order” throughout the manuscript (lines 80, 88, 194, 276) to improve readability and decrease word count. Along the same lines, it would be helpful to broaden the readership and interest of the article if authors could introduce the role, albeit briefly, of all genes and loci manipulated in experiments performed; for example, *retS*, *tssB1*, several others can be found throughout the manuscript that deserve improved descriptions.

These changes have been done, and a brief description of *retS* and *tssB1* has been added to the corresponding figure legend (lines 830-832).

In figure 2E indicate whether the difference in survival of *Pa* Δ tssM1 compared to *PA* Δ gacA subjected to attach by *Ab* wt is statistically significant. Similarly, indicate where the differences shown in Figure 2D supplement are statistically significant.

The requested comparisons have been performed and included into the figures.

The colored lines of the CPRG-based lysis assays are very thin and difficult to discern in the graph and the shading made this worse. I could not distinguish any of the strains plotted in graphs 3B-E, 4A and 4B. The figure legend does not explain the reason for the shadings; i.e. whether they have anything to

do with error range or whatever maybe highlighted. Please indicate what OD 575nm is representative of along the Y axis of these analogous graphs for clarity. How many replicates of these experiments were conducted? Could these experimental results be presented differently? For example by calculating and graphing out the slopes; or perhaps in table format?

The results were displayed differently by showing accumulated OD over a 10h period, and the lysis curves have been moved to supplementary materials. The legend of the supplementary materials has been corrected to indicate that the shading represents the standard deviation. The Y axis has been relabeled as “Cumulative OD_{575nm} (cell lysis)”. The experiments were performed in triplicate and this has been added to the figure legend.

Please add sufficient details to all figure legends throughout the manuscript; they are far too brief and succinct and devoid of information to be helpful.

Figure legends have been expanded.

Figure 6A: Why are 3 colonies highlighted in the inset? This figure is missing WT struck on LB-NaCl, WT struck on LB no NaCl, and $\Delta oprF$ struck on LB-NaCl for CFU size comparisons. Please add the missing controls (4 quadrants).

A new image has been included in Fig 6A with the requested information. There was no particular reason for having 3 colonies in the inset, it was just a visualization of the heterogeneity of colony size.

Figure 6B: The graph is un-interpretable: see previous comments about similar previous graphs, line thickness, colors, shading – none explained in the figure legend. Six strains appear to have been assayed; at least nine plots can be (barely) distinguished when zooming in.

Color code has been modified. The parental strain is displayed in black. In yellow is the $\Delta oprF$ deletion that displays a growth defect under these conditions. In shades of blue/purple are 4 suppressor colonies, all of which behave in a similar manner. The figure legend has been corrected to indicate the meaning of the shading.

Do the authors have an explanation for the large number of $\Delta oprF$ clones grown on LB w/o NaCl of which 4 revealed to harbor 413 bp deletions within *gacS*? Do the authors expect that all several 100 CFUs would harbor the same deletion? Any speculations on the odds of this event and what it may mean? On the other hand, such considerations would be best reserved for a manuscript that clarifies the “genetic linkage” between *gacS* and *oprF*.

It may well be that all 4 sequenced isolates descend from the same suppressor lineage and harbor the same deletion. A similar deletion was observed often in different experiments but the exact frequency has not been quantified. Indeed, a more in-depth exploration of the frequency of this phenomenon can be included in a follow-up study on the genetic linkage between *gacS* and *oprF*.

Why did the authors not pursue examination of the $\Delta gacS$ strain antibiotic resistance and instead chose the *oprF* deletion strain which has a demonstrated growth defect?

We now performed an experiment on a $\Delta gacA$ strain that lacks the effector protein of the *gacS-gacA* two component system. This strain displays an even stronger phenotype than the others that are

included in this study, suggesting that there may be other genes from the GacA regulon that contribute to the observed phenotype. This will also be the subject of follow-up studies.

References

1. Hersch, S. J. *et al.* Envelope stress responses defend against type six secretion system attacks independently of immunity proteins. *Nat Microbiol* **5**, 706–714 (2020).
2. Dong, T. G., Ho, B. T., Yoder-Himes, D. R. & Mekalanos, J. J. Identification of T6SS-dependent effector and immunity proteins by Tn-seq in *Vibrio cholerae*. *Proc Natl Acad Sci U S A* **110**, 2623–2628 (2013).
3. Monzingo, A. F., Marcotte, E. M., John Hart, P. & Robertus, J. D. Chitinases, chitosanases, and lysozymes can be divided into procaryotic and eucaryotic families sharing a conserved core. *Nat Struct Biol* **3**, 133–140 (1996).
4. Saito, J. I. *et al.* Crystal Structure of Chitosanase from *Bacillus circulans* MH-K1 at 1.6-Å Resolution and Its Substrate Recognition Mechanism. *Journal of Biological Chemistry* **274**, 30818–30825 (1999).
5. Ringel, P. D., Hu, D. & Basler, M. The Role of Type VI Secretion System Effectors in Target Cell Lysis and Subsequent Horizontal Gene Transfer. *Cell Rep* **21**, 3927–3940 (2017).
6. Tang, M.-X. *et al.* Abiotic factors modulate interspecies competition mediated by the type VI secretion system effectors in *Vibrio cholerae*. *ISME J* 1–11 (2022) doi:10.1038/s41396-022-01228-5.
7. George, M., Narayanan, S., Tejada-Arranz, A., Plack, A. & Basler, M. Initiation of H1-T6SS dueling between *Pseudomonas aeruginosa*. *mBio* **15**, (2024).
8. Ting, S. Y. *et al.* Discovery of coordinately regulated pathways that provide innate protection against interbacterial antagonism. *Elife* **11**, (2022).

REVIEWERS' COMMENTS

Reviewer #1 (Remarks to the Author):

The authors have adequately addressed all my concerns. The findings of this manuscript provide additional insights that advance our understanding of bacterial non-specific defense mechanisms against T6SS-mediated attacks.

Reviewer #2 (Remarks to the Author):

The authors have made an effort to address the comments that arose during the first round of revision. The manuscript has been improved, and I am happy with most of it. I only have a few additional very minor comments:

Line 31: H3-T6SS also delivers effectors involved in common goods.

A sentence indicating the roles of the H3-T6SS in iron acquisition and biofilm formation has been added.

Line 46: “composed of” instead of “composed by”

Fixed.

Line 84: misspelling in the word “cholerae”

Fixed.

Line 111-112: “compared” appears twice very closely. There are other examples with words like “important”, “resistance”, “such as” throughout the manuscript.

These repetitions have been modified throughout the manuscript.

Line 112/196: parental strain. Line 247: antimicrobial compound (the nouns were missing)

Fixed.

Line 178: “or” instead of “nor”

Fixed.

The authors should be consistent with the word “wild-type”, sometimes it is written “wt”, others WT, on other occasions “wild type” and lastly “wild-type”

Fixed.

Line 219: “without” instead “no”

Fixed.

Line 243 – space between “dtssM1” and “strain”

Fixed.

Line 464: bacteria “were” spun down

Fixed.

Line 468: similarly “to” earlier instead of “as”

Fixed.

Line 538: repetitive “plasmid-encoded” and “pME3856” plasmid

Fixed.

The structure “or not” should be substituted with “do not have” (lines 739, 742, 792, 800, 822, ...)

Fixed.

Line 752: genes “encoding proteins” important for ...

Fixed.

Figure legend 9: it is a very long sentence that could benefit from be split at least in 2.

Fixed.